# Dynamic Multi-sample Mixup with Gradient Exploration for Open-set Graph Anomaly Detection

**Caiyang Yu**[1]    **Wei Ju**[1*]    **Haixin Wang**[2]    **Yifan Wang**[3]    **Ziyue Qiao**[4]

[1]College of Computer Science, Sichuan University of China
[2]Computer Science Department, University of California, Los Angeles
[3]School of Artificial Intelligence and Data Science, University of International Business and Economics
[4]School of Computing and Information Technology, Great Bay University
`{yucy324, ziyuejoe}@gmail.com`
`juwei@scu.edu.cn`
`whx@cs.ucla.edu`
`yifanwang@uibe.edu.cn`

## Abstract

This paper studies the problem of open-set graph anomaly detection, which aims to generalize a graph neural network (GNN) trained with a small number of both normal and abnormal nodes to detect unseen anomalies different from training anomalies during inference. This problem is highly challenging due to both the data scarcity of unseen anomalies and the label scarcity for training nodes. Towards this end, we propose a novel approach named Dynamic Multi-sample Mixup with Gradient Exploration (**DEMO**) for open-set graph anomaly detection. The core of our proposed **DEMO** is to leverage a dynamic framework to adapt the optimization procedure with high generalizability. In particular, our **DEMO** first adaptively fuses multiple seen nodes to simulate the unseen anomalies, which expands the decision boundary for the detection model with enhanced generalizability. Moreover, we dynamically adjust sample weights based on their energy gradients to prioritize uncertain and informative nodes, ensuring a robust optimization procedure. To further address both label scarcity and severe class imbalance, we maintain a memory bank of historical records to guide the pseudo-labeling process of unlabeled nodes. Extensive experiments on various benchmark datasets validate the superiority of the proposed **DEMO** in comparison to various baselines. Code is available at `https://github.com/yucy324/DEMO`.

## 1 Introduction

Graph anomaly detection (GAD) focuses on identifying rare or malicious patterns in graph-structured data that significantly deviate from expected behaviors (Akoglu et al., 2015; Ma et al., 2021; Tang et al., 2023), such as fraudulent transactions in financial networks (Kim et al., 2024) or stealthy attacks in Internet-of-Things (IoT) systems (Wu et al., 2021). Compared to traditional Euclidean space-based anomaly detection, GAD requires the joint modeling of node attributes and topological structure, where anomalies typically manifest as attribute distributional shifts coupled with structural inconsistencies. This dual dependency introduces unique technical challenges: anomalies often exhibit high structural diversity and strong context dependence, making them inherently more difficult to define and detect (Vaska et al., 2022). As such, designing GAD approaches that are both effective and generalizable remains a challenging and open research problem.

Current GAD methods predominantly rely on unsupervised and semi-supervised learning paradigms. Unsupervised approaches exploit topological structures and attribute statistics of graphs to identify anomalies in the absence of labeled data. These methods typically estimate anomaly scores based on various principles (Ding et al., 2019; Liu et al., 2021; Ni et al., 2025), offering strong generalizability

---

*Corresponding author.

across diverse graph domains. However, these frameworks suffer from limited precision due to the absence of semantic guidance on anomaly properties. In contrast, semi-supervised frameworks (Dong et al., 2025a; Chen et al., 2024; Ding et al., 2021a) leverage scarce labeled normal or anomalous samples to guide learning processes, thereby achieving more discriminative representations and enhanced detection performance. These methods typically incorporate consistency regularization (Chen et al., 2024), generative objectives (Zhang et al., 2022a), and graph-based data augmentation (Liu et al., 2022) strategies to improve model capabilities from limited supervision. Despite technical diversity among existing approaches, they universally operate under a closed-set assumption, presuming training data fully represents all possible anomaly types or their underlying distributions, a premise that critically undermines their practicality in real-world graph analytics scenarios.

In this paper, we focus on an under-explored yet critically important problem: open-set graph anomaly detection (open-set GAD). The goal of this task is to train a GAD model using a limited number of labelled nodes, with the dual objective of detecting both seen anomalies and unseen ones that exhibit behavior patterns significantly distinct from training anomalies and absent from the training data. Achieving this goal is non-trivial, as it entails addressing several interrelated and fundamental challenges in graph-based settings. **I)** *How to generalize knowledge from limited seen anomalies to detect unseen anomalies?* In practice, the training set for open-set GAD often has two critical limitations: the number of anomaly classes is small, and the diversity of these classes is insufficient. To detect unseen anomalies, it is essential to fully extract and leverage all useful information from the few seen anomalies available. However, existing methods (e.g., (Wang et al., 2023b)) fail to tap into this potential, leading to a critical gap in using such supervision to support unseen anomaly detection. **II)** *How to perform effective graph anomaly detection under limited labels and severe class imbalance?* Open-set GAD faces two overlapping constraints: extremely scarce labeled data and severe class imbalance (normal nodes dominate the graph, while labeled/unlabeled anomalies are extremely rare). Existing semi-supervised GAD methods often struggle with this combination: they tend to overfit the dominant normal class, resulting in biased decision boundaries. These biased boundaries not only perform poorly on rare anomalies but also fail to generalize to unseen anomalies.

To address the aforementioned challenges, we propose a novel approach named Dynamic Multi-sample Mixup with Gradient Exploration (named **DEMO**) for open-set GAD. The goal of **DEMO** is to establish a dynamical adaptive training framework that improves generalization to unseen anomalies and enhances anomaly detection performance under limited supervision. Given the scarcity and homogeneity of anomaly classes in the training set, **DEMO** first adaptively fuses multiple seen anomaly samples to generate synthesized nodes with enriched representations, approximating unseen anomalies to drive the learning of broader decision boundaries. Moreover, while augmenting the training set with synthesized anomalies is beneficial, not all samples contribute positively to model generalization. Consequently, we employ an energy gradient-driven feedback mechanism to evaluate and re-weight each sample dynamically. This ensures the model prioritizes the optimization of latent uncertain samples crucial for generalization, effectively guiding the training process. Finally, to directly address the dual challenges of limited labels and severe class imbalance, **DEMO** uses a memory bank to guide pseudo-labeling with adaptive, class-specific confidence thresholds, thereby mitigating the resulting training bias. Extensive experiments on diverse benchmark datasets demonstrate **DEMO**'s consistently superior performance against state-of-the-art baselines, effectively validating its robustness and effectiveness in challenging open-set scenarios.

The contribution of this paper is summarized as follows: ❶ *Problem Connection.* We present a new perspective that connects open-set recognition with GAD under label scarcity, emphasizing the need to generalize beyond seen anomalies and revisit GAD through the lens of open-set detection. ❷ *Novel Framework.* We propose a novel framework named **DEMO**, which leverages a multi-sample mixup strategy and energy gradient-based feedback mechanism to fully exploit limited labeled data. Furthermore, **DEMO** incorporates a memory bank to mitigate label scarcity by guiding the pseudo-labeling process of unlabeled nodes. ❸ *Comprehensive Validation.* Extensive experiments on multiple real-world benchmark datasets and under various challenging evaluation settings validate the superiority of **DEMO** over a diverse range of state-of-the-art GAD baselines.

## 2 RELATED WORK

**Graph Anomaly Detection.** Existing graph anomaly detection (GAD) methods generally fall into three categories: unsupervised (Huang et al., 2023; Dong et al., 2025b; Qiao & Pang, 2023), semi-supervised (Gao et al., 2023; Huang et al., 2022; Dong et al., 2025a), and supervised (Ding et al., 2021b) methods. Unsupervised GAD does not rely on labeled anomalies, which identify outliers by modeling the graph's inherent structural and attribute distributions (Ding et al., 2019; Fan et al., 2020). However, such methods often struggle to disentangle meaningful information from noise in the latent space. To improve discriminative ability, recent work incorporates diffusion-guided refinement of latent representations and content-preserving constraints, enhancing anomaly separability beyond what pure reconstruction can achieve (Li et al., 2024). Additionally, some unsupervised approaches integrate contrastive learning (Duan et al., 2023b;a) or generative modeling (Chen et al., 2020b) to bolster the robustness of learned representations. In contrast, semi-supervised GAD methods utilize a limited number of labeled anomalies to guide the learning process and enhance detection accuracy. Common strategies include consistency training to enforce prediction stability under perturbations (Chen et al., 2024), as well as graph-specific augmentations that improve the model's ability to generalize from scarce supervision (Liu et al., 2022). Fully supervised GAD methods, on the other hand, assume comprehensive labels. Recent work has focused on complex generalization challenges, such as meta-learning for few-shot detection (Meta-GDN) (Ding et al., 2021b), cross-domain GAD (ACT) (Wang et al., 2023a), and generalist GAD (ARC (Liu et al., 2024), AnomalyGMF (Qiao et al., 2025)) aiming to unify performance across diverse graphs and anomaly types. However, most existing approaches, whether semi-supervised or fully supervised, assume a consistent anomaly distribution between training and testing, ignoring real-world variability and thereby limiting their ability to generalize to unseen anomaly types. To address this limitation, we propose a novel open-set GAD framework explicitly tailored for detecting diverse and previously unseen anomalies.

**Open-set Classification.** Open-set classification addresses a practical challenge where the model must not only accurately classify inputs from known categories but also identify instances originating from previously unseen classes (Yang et al., 2024; Wang et al., 2023b; Bisgin et al., 2024). Existing open-set classification approaches are mainly divided into discriminative (Luke et al., 2020; Chen et al., 2020a; Xu, 2024) and generative paradigms (Bao et al., 2023). Discriminative methods aim to learn well-separated decision boundaries for known classes while reserving ambiguous regions in the feature space to detect unknowns. In contrast, generative methods attempt to simulate the behavior of unknown classes by explicitly synthesizing samples, often combining generative models with discriminative classifiers in an adversarial framework. In the open-set GAD, NSReg (Wang et al., 2023b) is one of the few existing approaches that has shown promising results. It adopts a discriminative framework by introducing a regularization constraint to enforce compact semantic representations of normal nodes, thereby reducing overfitting to anomalies. However, its core idea focuses on constraining structural relations among normal nodes to enforce a strict decision boundary, while overlooking the role of anomaly nodes in shaping the model's behavior. Therefore, our study emphasizes the significance of anomaly samples and achieves a diverse representation through seen anomalies to enhance the generalization ability of the model.

## 3 THE PROPOSED APPROACH

**Problem Definition.** We consider the task of open-set GAD on attributed graphs, where only a small portion of anomalies are labeled during training and novel anomaly types may emerge at inference. Let $\mathcal{G} = (\mathcal{V}, \mathcal{E}, \mathbf{X})$ denote a graph with node set $\mathcal{V}$, edge set $\mathcal{E}$, and feature matrix $\mathbf{X} \in \mathbb{R}^{|\mathcal{V}| \times d}$. The node set consists of a dominant group of normal nodes $\mathcal{V}_n$ and a minority of anomalous nodes $\mathcal{V}_a$, *i.e.*, $\mathcal{V} = \mathcal{V}_n \cup \mathcal{V}_a$, where $|\mathcal{V}_a| \ll |\mathcal{V}_n|$. During training, the labeled training nodes $\mathcal{V}^{\text{train}}$ only cover a partial set of anomaly classes. To formalize this, we partition $\mathcal{V}_a$ into seen anomalies $\mathcal{V}_a^{\text{seen}}$ and unseen anomalies $\mathcal{V}_a^{\text{unseen}}$. The objective is to learn a detection model $\phi : (\mathcal{G}, \mathcal{V}) \to [0, 1]$ that assigns high anomaly scores to both $\mathcal{V}_a^{\text{seen}}$ and $\mathcal{V}_a^{\text{unseen}}$, while suppressing scores for $\mathcal{V}_n$. Formally, for all $v_a \in \mathcal{V}_a$ and $v_n \in \mathcal{V}_n$, the model satisfies: $\phi(\mathcal{G}, v_a) \gg \phi(\mathcal{G}, v_n)$.

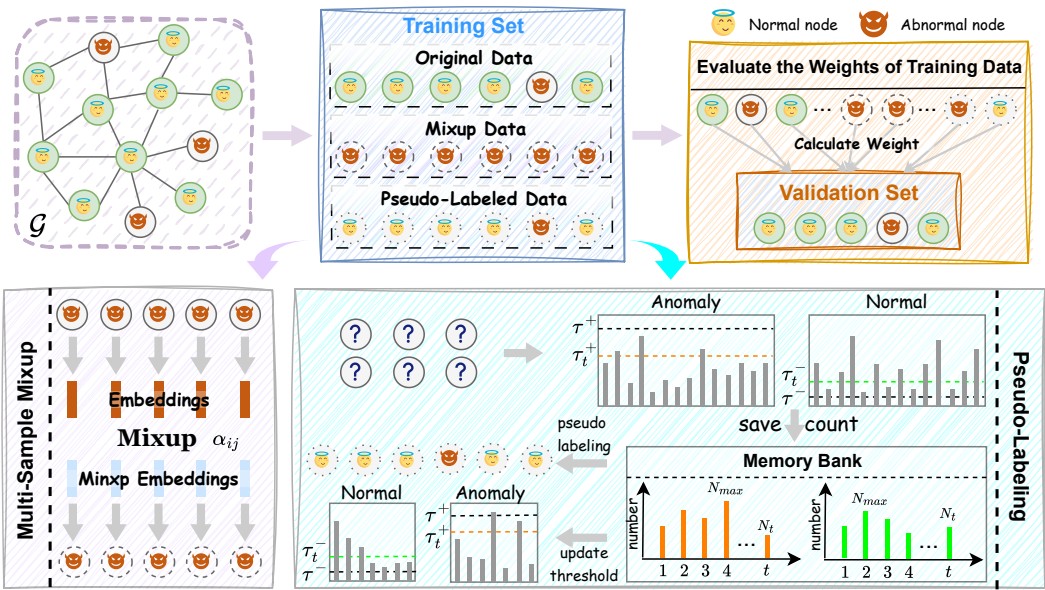

Figure 1: An overview of the proposed **DEMO**. **DEMO** first expands the training data through two parallel augmentation techniques. **Multi-Sample Mixup** generates new synthetic anomalies, while **Pseudo-Labeling** assigns labels to reliable unlabeled nodes. This augmented training data, comprising original, mixup, and pseudo-labeled data, then proceeds to a dynamic weighting stage.

## 3.1 FRAMEWORK OVERVIEW

Our study addresses the open-set GAD problem, aiming to detect both seen and unseen anomalies using limited labeled nodes. This involves two key challenges: (1) *Generalization to Unseen Anomalies*. (2) *Label Scarcity and Class Imbalance*. To tackle both challenges, we propose a novel framework **DEMO** that enhances the generalization of GNN-based detectors under minimal supervision. First, **DEMO** adaptively fuses multiple seen anomalies to synthesize diverse nodes that simulate unseen anomaly classes, thereby expanding the model's decision boundaries for improved generalization. Second, an energy gradient-driven feedback mechanism dynamically adjusts sample weights, enabling the model to focus on uncertain or ambiguous instances and improve generalization. Finally, to mitigate bias caused by severe class imbalance and limited anomaly supervision, we introduce a memory bank of historical records to adaptively update class-specific confidence thresholds and guide the pseudo-labeling of unlabeled nodes. The overall architecture of **DEMO** is illustrated in Figure 1, with detailed functionalities of each component elaborated in subsequent sections.

## 3.2 DYNAMIC MULTI-SAMPLE MIXUP FOR DECISION BOUNDARY EXPANSION

In the context of open-set GAD, existing methods (Wang et al., 2023b; Zhu et al., 2024; Isaac-Medina et al., 2024) struggle to capture the diversity of anomalies due to the limited quantity and class variety of anomalies in the training data. Consequently, it becomes crucial to introduce diversified anomaly samples to enhance the model's generalization. To this end, **DEMO** employs a dynamic multi-sample mixup strategy that generates synthesized anomalies with diverse representations, effectively prompting the model to learn broader and more robust decision boundaries. The core innovation lies in adaptively fusing multiple seen anomalies to generate more challenging and diverse anomaly representations, which not only approximate the distribution of potential unseen anomalies but also systematically encourage the model to learn more robust decision boundaries, thereby enhancing generalization to previously unseen patterns.

**Definition 3.1** (Multi-sample Mixup) *Given the seen anomaly node set $\mathcal{V}_a^{train} = \{v_1^{train}, v_2^{train}, \cdots, v_N^{train}\}$, which are encoded into embeddings $\mathbf{Z}_a^{train} = \{z_1^{train}, z_2^{train}, \cdots, z_N^{train}\}$. For each original anomaly $z_i^{train} \in \mathbf{Z}_a^{train}$, we generate a corresponding synthetic representation $\hat{z}_i$. This*

*synthetic sample is defined as a dynamic mix of all seen anomalies' embeddings:* $\hat{z}_i = \sum_{j=1}^{N} \alpha_{ij} z_j^{train}$, *where* $\sum_{j=1}^{N} \alpha_{ij} = 1$ *and* $\alpha_{ij} \in [0, 1]$.

Notably, the mixing weights $\alpha_{ij}$ are assigned based on the feature similarity between embeddings (Zhang et al., 2022b), defined through the following normalized form:

$$\alpha_{ij} = \frac{\exp\left(\mathcal{S}\left((z_i^{\text{train}})^\top \mathbf{w}_m, (z_j^{\text{train}})^\top \mathbf{w}_n\right)\right)}{\sum_k \exp\left(\mathcal{S}\left((z_i^{\text{train}})^\top \mathbf{w}_m, (z_k^{\text{train}})^\top \mathbf{w}_n\right)\right)}, \tag{1}$$

with $\mathcal{S}$ representing a feature similarity function (*e.g.*, inner product or cosine similarity), $\mathbf{w}_m$ and $\mathbf{w}_n$ are learnable weights. Since highly similar samples are more confusable to the model, assigning larger weights to them encourages the synthesized representations to stay in more ambiguous regions of the feature space. Next, we provide a theoretical justification showing that the synthesized samples can still retain high similarity with the original ones. The detailed proof is presented in the Appendix C.

**Theorem 3.1** *Assume the inner-product similarity* $\mathcal{S}(z_i, z_j) = z_i^\top z_j$ *between original samples* $z_i^{train}$ *and* $z_j^{train}$ *is higher than their average similarity to all training anomalies, i.e.,* $\mathcal{S}(z_i^{train}, z_j^{train}) \geq \frac{1}{N} \sum_{k=1}^{N} \mathcal{S}(z_i^{train}, z_k^{train})$, *then the synthesized sample* $\hat{z}_i$ *satisfies:*

$$\mathcal{S}(\hat{z}_i, z_j) \geq \mathcal{S}(z_i, z_j) - \epsilon, \tag{2}$$

*where* $\epsilon = \sum_{k=1}^{N} \alpha_{ik} |\mathcal{S}(z_k, z_j) - \mathcal{S}(z_i, z_j)|$. *That is, the similarity between the mixed sample* $\hat{z}_i$ *and* $z_j$ *is guaranteed to be no less than the original value minus a small perturbation* $\epsilon$, *thereby preserving their intrinsic confusability.*

Furthermore, to avoid degenerate cases where the synthesized representation is overly biased toward the original sample $z_i$, we introduce a diversity regularization term to suppress its dominant influence:

$$\mathcal{L}_{\text{div}} = -\frac{1}{N} \sum_i \left\| \frac{(z_i^{\text{train}})^\top \mathbf{w}_m}{\left\|(z_i^{\text{train}})^\top \mathbf{w}_m\right\|_2} - \frac{(z_i^{\text{train}})^\top \mathbf{w}_n}{\left\|(z_i^{\text{train}})^\top \mathbf{w}_n\right\|_2} \right\|^2. \tag{3}$$

Upon synthesizing diverse anomalies, we further enhance the discriminative power of the model through consistency learning. The final loss function integrates both the diversity enhancement term (with a hyperparameter $\lambda_{\text{div}}$) and the consistency learning term as follows:

$$\mathcal{L}_{\text{mix}} = \mathcal{L}_{\text{cons}} + \lambda_{\text{div}} \mathcal{L}_{\text{div}}, \tag{4}$$

where $\mathcal{L}_{\text{cons}}$ is the consistency learning loss between the original anomaly embeddings $x_i^{\text{train}} \in \mathbf{X}$ and their projected representations $z_i^{\text{train}}$, calculated using the Mean Squared Error (MSE).

### 3.3 GRADIENT EXPLORATION FOR GENERALIZABLE OPTIMIZATION

While the multi-sample mixup strategy creates a more diverse set of anomalies, not all samples in this expanded training set (both original and synthesized) are equally important for the model's generalization. Simply treating all samples equally is suboptimal, as some may be highly informative boundary cases while others might be redundant or even introduce noise. To address this, we propose an energy gradient-driven mechanism (Chen et al.; Han et al., 2022) that dynamically weights each sample based on its quantified impact on validation performance, thereby focusing optimization on the most critical nodes for more effective learning.

We define the energy of a node as a measure of predictive uncertainty, formulated as $E_\theta(v_i) = -\log \sum \exp(z_i)$, where $z_i$ denotes the predicted logits of node $v_i$. A lower energy indicates higher confidence, typically associated with normal nodes, while a higher energy suggests potentially anomalous or ambiguous behavior. To characterize how node energy affects generalization, we analyze model parameter responses to minor energy perturbations. The response direction is determined by the energy gradient to model parameters, with magnitude governed by the Hessian matrix $H_{\hat{\theta}}$:

$$\mathcal{I}_{\hat{\theta}}(v_i) = -H_{\hat{\theta}}^{-1} \nabla_\theta E_\theta(v_i). \tag{5}$$

The Hessian matrix is evaluated at optimal parameters, quantifying sensitivity to local perturbations. However, parameter shifts alone cannot determine generalization benefits. We further evaluate their

impact on validation loss by computing the inner product between validation loss gradients and energy gradient directions:

$$\mathcal{I}_{v_j^{\text{val}}}(v_i) = -\nabla_\theta \mathcal{L}\left(v_j^{\text{val}}, y_j^{\text{val}}; \hat{\theta}\right)^\top H_{\hat{\theta}}^{-1} \nabla_\theta E_\theta(v_i), \tag{6}$$

where $\mathcal{L}(\cdot)$ denotes the binary cross-entropy loss, $v_j^{\text{val}}$ and $y_j^{\text{val}}$ are the nodes of the validation set and the corresponding categories. Averaging over all validation nodes yields the average influence $T_{\text{val}}(v_i)$ of training node $v_i$, which determines the adaptive weight coefficient:

$$\beta_{v_i} = -\frac{T_{\text{val}}(v_i)}{\max_{v_k \in \mathcal{V}^{\text{train}}} |T_{\text{val}}(v_k)|}. \tag{7}$$

Finally, we incorporate these weights into the energy-aware objective and define the overall loss as:

$$\mathcal{L}_{\text{energy}} = \frac{1}{n} \sum_{i=1}^{n} \left[\mathcal{L}(v_i, y_i; \theta) + \lambda_{\text{eng}} \beta_{v_i} \cdot E_\theta(v_i)\right], \tag{8}$$

where $\lambda_{\text{eng}}$ is a balance coefficient. The core idea of this mechanism is that when $T_{\text{val}}(v_i) > 0$, reinforcing the energy guidance of node $v_i$ can effectively reduce validation error. Such nodes are often boundary cases that may resemble unseen anomalies. Conversely, a negative influence suggests that the sample may impair generalization, and its contribution should be suppressed.

## 3.4 RELIABLE PSEUDO-LABELING WITH HISTORICAL GUIDANCE

After effectively leveraging labeled data to enhance the model's generalization to unseen anomalies, another crucial objective focuses on generating high-quality pseudo-labels for unlabeled samples to further alleviate the training bottleneck caused by label scarcity. Traditional pseudo-labeling methods employing fixed thresholds often fail to account for the dynamic evolution of model prediction behaviors during training (Xie et al., 2020; Guo & Li, 2022). Meanwhile, dynamic thresholding strategies (Zhang et al., 2021; Wang et al., 2023c; Chen et al., 2023) perform poorly in imbalanced binary-class GAD due to their class-agnostic design, which often overlooks anomalies. To address this, we propose a class-aware threshold adaptation scheme guided by historical records, enabling reliable pseudo-label generation.

In detail, our method implements class-specific dynamic threshold adaptation via a memory bank that incorporates historical information. Let $\mathcal{T}_t^c$ denote the set of samples selected for class $c$ ($c \in \{0(\text{normal}), 1(\text{anomaly})\}$) at the $t$-th iteration, with its size $N_t^c = |\mathcal{T}_t^c|$ stored in the memory bank to track distribution evolution. By computing the historical peak $N_{\max}^c = \max_{1 \leq k \leq t} N_k^c$, the model dynamically captures the historical optimal coverage for class $c$. Based on the ratio between the current selection count $\sigma_t(c)$ and $N_{\max}^c$, the model dynamically adjusts the threshold $\rho_t(c) = \sigma_t(c)/N_{\max}^c$, where $\sigma_t(c)$ is determined by the following condition:

$$\sigma_t(c) = \sum_{i=1}^{M} \mathbb{I}\left[c = 1 \wedge \hat{p}_t(v_i) \geq \tau_t^+ \vee c = 0 \wedge \hat{p}_t(v_i) \leq \tau_t^-\right], \tag{9}$$

where $M$ is the number of unlabeled samples, $\hat{p}_t(v_i)$ is the predicted probability of node $v_i$ at iteration $t$, and $\tau_t^+$ and $\tau_t^-$ represents the dynamic thresholds for anomaly and normal classes, respectively. The final asymmetric threshold update mechanism is formulated as:

$$\tau_t^{+/-} = \begin{cases} \rho_t(\text{c}) \cdot \tau^+, & c = \text{anomaly}, \\ \tau^- \cdot (2 - \rho_t(\text{c})), & c = \text{normal}, \end{cases} \tag{10}$$

where $\tau^+$ and $\tau^-$ denote predefined anomaly and normal class thresholds, with $\tau^+ + \tau^- = 1$. This strategy ensures that anomaly thresholds $\tau_t^+$ progressively increase with $\rho_t(\text{anomaly})$ to enhance minority-class sensitivity, while normal thresholds $\tau_t^-$ decrease through the nonlinear term $(2 - \rho_t(\text{normal}))$, improving robustness against majority-class dominance.

## 3.5 SUMMARIZATION

In practice, we dynamically adjust the class-specific thresholds during training and select unlabeled samples. The overall training objective is defined as:

$$\mathcal{L} = \mathcal{L}_{\text{energy}} + \lambda_{\text{mix}} \mathcal{L}_{\text{mix}} + \lambda_{\text{un}} \mathcal{L}_{un}. \tag{11}$$

Table 1: AUC-ROC and AUC-PR on three small-scale datasets. The best performance is boldfaced, with the second-best underlined.

| Datasets | Photo | | Computers | | CS | |
|---|---|---|---|---|---|---|
| Metrics | AUC-ROC | AUC-PR | AUC-ROC | AUC-PR | AUC-ROC | AUC-PR |
| ANOMALOUS | $0.5574_{\pm0.012}$ | $0.0879_{\pm0.003}$ | $0.5737_{\pm0.016}$ | $0.1693_{\pm0.003}$ | $0.2997_{\pm0.017}$ | $0.1634_{\pm0.015}$ |
| DOMINANT | $0.4716_{\pm0.028}$ | $0.0837_{\pm0.009}$ | $0.5450_{\pm0.015}$ | $0.1644_{\pm0.008}$ | $0.4029_{\pm0.012}$ | $0.1886_{\pm0.024}$ |
| AnomalyDAE | $0.4179_{\pm0.032}$ | $0.0770_{\pm0.007}$ | $0.5658_{\pm0.011}$ | $0.1723_{\pm0.007}$ | $0.3978_{\pm0.009}$ | $0.1864_{\pm0.008}$ |
| GAAN | $0.4346_{\pm0.014}$ | $0.0710_{\pm0.003}$ | $0.5595_{\pm0.029}$ | $0.1796_{\pm0.011}$ | $0.4646_{\pm0.026}$ | $0.2111_{\pm0.017}$ |
| CoLA | $0.5618_{\pm0.008}$ | $0.0989_{\pm0.006}$ | $0.4897_{\pm0.010}$ | $0.1472_{\pm0.006}$ | $0.4353_{\pm0.014}$ | $0.2029_{\pm0.015}$ |
| CONAD | $0.4763_{\pm0.037}$ | $0.0862_{\pm0.011}$ | $0.5445_{\pm0.023}$ | $0.1619_{\pm0.015}$ | $0.4028_{\pm0.006}$ | $0.1886_{\pm0.009}$ |
| CONSISGAD | $\underline{0.8668}_{\pm0.021}$ | $\underline{0.5987}_{\pm0.002}$ | $0.6250_{\pm0.017}$ | $0.3572_{\pm0.009}$ | $0.7178_{\pm0.034}$ | $0.5271_{\pm0.019}$ |
| GGAD | $0.7976_{\pm0.032}$ | $0.5677_{\pm0.004}$ | $0.7210_{\pm0.043}$ | $0.4529_{\pm0.014}$ | $\underline{0.9081}_{\pm0.029}$ | $\underline{0.8198}_{\pm0.022}$ |
| TAM | $0.6045_{\pm0.015}$ | $0.1084_{\pm0.003}$ | $0.4432_{\pm0.014}$ | $0.1355_{\pm0.010}$ | $0.6398_{\pm0.011}$ | $0.3542_{\pm0.013}$ |
| OCGNN | $0.6279_{\pm0.007}$ | $0.1323_{\pm0.007}$ | $0.5049_{\pm0.022}$ | $0.1505_{\pm0.013}$ | $0.7819_{\pm0.031}$ | $0.4926_{\pm0.028}$ |
| ANO-S | $0.5730_{\pm0.011}$ | $0.1097_{\pm0.006}$ | $0.4628_{\pm0.004}$ | $0.1392_{\pm0.005}$ | $0.8380_{\pm0.025}$ | $0.6401_{\pm0.014}$ |
| DOM-S | $0.5785_{\pm0.063}$ | $0.1107_{\pm0.013}$ | $0.4488_{\pm0.007}$ | $0.1330_{\pm0.003}$ | $0.8445_{\pm0.013}$ | $0.6382_{\pm0.019}$ |
| SpaceGNN | $0.8030_{\pm0.005}$ | $0.5271_{\pm0.009}$ | $\underline{0.8296}_{\pm0.032}$ | $\underline{0.6439}_{\pm0.019}$ | $0.7784_{\pm0.042}$ | $0.6587_{\pm0.031}$ |
| NSReg | $0.8360_{\pm0.012}$ | $0.4777_{\pm0.018}$ | $0.7403_{\pm0.021}$ | $0.5437_{\pm0.013}$ | $0.9032_{\pm0.035}$ | $0.8115_{\pm0.016}$ |
| GNN+OpenMax | $0.7618_{\pm0.104}$ | $0.3942_{\pm0.063}$ | $0.6713_{\pm0.052}$ | $0.3942_{\pm0.095}$ | $0.8213_{\pm0.133}$ | $0.7559_{\pm0.049}$ |
| **DEMO** | $\mathbf{0.9023}_{\pm0.009}$ | $\mathbf{0.6330}_{\pm0.006}$ | $\mathbf{0.8439}_{\pm0.024}$ | $\mathbf{0.6458}_{\pm0.013}$ | $\mathbf{0.9448}_{\pm0.019}$ | $\mathbf{0.8857}_{\pm0.010}$ |

where $\lambda_{\mathrm{mix}}$ and $\lambda_{\mathrm{un}}$ are the balancing coefficients for the mixup loss and the unlabeled loss, respectively. $\mathcal{L}_{\mathrm{un}}$ denotes the binary cross-entropy loss applied to pseudo-labeled samples. The detailed training procedure of **DEMO** is provided in the Appendix B.

## 4 EXPERIMENTS

### 4.1 EXPERIMENTAL SETTINGS

**Datasets.** To comprehensively evaluate the performance of **DEMO** on the open-set GAD task, we conduct experiments on six real-world graph datasets that vary in scale and domain characteristics. Since no existing graph benchmarks are explicitly designed for open-set scenarios, we simulate by partitioning normal and anomalous nodes based on their non-uniform class distributions. Following (Wang et al., 2023b), we define normal nodes as majority classes and partition anomalies based on class proportions: (1) For small-scale datasets with multiple classes, such as *Photo*, *Computers*, and *CS* (Shchur et al., 2018), we treat categories with fewer than 5% of the total nodes as anomaly classes (with at least two anomaly classes retained); (2) For large-scale datasets, including *Yelp* (Rayana & Akoglu, 2015), *ogbn-arxiv* (Mikolov et al., 2013), and *ogbn-mag* (Wang et al., 2020), we apply a similar strategy based on class proportions to distinguish normal and anomalous nodes. Further implementation details are provided in the Appendix.

**Baselines.** In our experiments, we compare **DEMO** with 12 representative GAD methods, which are categorized into two groups: unsupervised and semi-supervised approaches. The unsupervised methods include ANOMALOUS (Peng et al., 2018), DOMINANT (Ding et al., 2019), AnomalyDAE (Fan et al., 2020), GAAN (Chen et al., 2020b), CoLA (Liu et al., 2021), and CONAD (Xu et al., 2022). The semi-supervised group includes ConsisGAD (Chen et al., 2024), GGAD (Qiao et al., 2024), TAM (Qiao & Pang, 2023), OCGNN (Wang et al., 2021), ANOMALOUS-Semi (ANO-S), DOMINANT-Semi (DMO-S), SpaceGNN (Dong et al., 2025a), and NSReg (Wang et al., 2023b). Notably, TAM, OCGNN, ANOMALOUS, and DOMINANT are originally unsupervised models. Following the design strategy in GGAD, we adapt them into semi-supervised variants to ensure fair comparisons under the same supervision setting. Finally, we add GNN+OpenMax (Bendale & Boult, 2016), which adapts the classic open-set classification framework to a GNN baseline.

**Implementation Details and Evaluation Metrics.** Following (Wang et al., 2023b), we adopt GraphSAGE as the backbone with 64 hidden units per layer. The model is optimized using the Adam optimizer with a learning rate of 0.001 and a weight decay of 0.0005. For training, we run

Table 2: AUC-ROC and AUC-PR on three large-scale datasets. The best performance is boldfaced, with the second-best underlined. 'OOM' indicates out-of-memory.

| Datasets | Metrics | ConsisGAD | GGAD | TAM | OGCNN | ANO-S | DOM-S | SpaceGNN | NSReg | **DEMO** |
|----------|---------|-----------|------|-----|-------|-------|-------|----------|-------|----------|
| Yelp | AUC-ROC | 0.6988 | 0.6613 | 0.5319 | 0.6410 | 0.6567 | 0.6506 | 0.6853 | 0.7015 | **0.7097** |
| | AUC-PR | 0.2970 | 0.2549 | 0.0977 | 0.1118 | 0.1076 | 0.1048 | 0.2916 | **0.3029** | 0.2238 |
| ogbn-arxiv | AUC-ROC | 0.6216 | 0.6007 | OOM | OOM | 0.4510 | 0.4505 | 0.6133 | 0.6182 | **0.6364** |
| | AUC-PR | 0.3148 | 0.2843 | OOM | OOM | 0.1463 | 0.1482 | 0.3301 | 0.3230 | **0.3329** |
| ogbn-mag | AUC-ROC | 0.4909 | OOM | OOM | OOM | OOM | OOM | 0.4626 | 0.4836 | **0.4967** |
| | AUC-PR | 0.0043 | OOM | OOM | OOM | OOM | OOM | 0.0043 | 0.0041 | **0.0054** |

200 epochs on small-scale datasets and 400 epochs on large-scale datasets. We select 50 anomalous nodes (from a single anomaly class) and 5% of normal nodes as the training set, and 30 anomalous nodes (same class as training) along with 1% of normal nodes as the validation set; all remaining nodes are used for testing. We set the hyperparameter $\lambda_{\text{div}}$ to 0.5, and conduct a sensitivity analysis on the loss weights $\lambda_{\text{eng}}$, $\lambda_{\text{mix}}$, and $\lambda_{\text{un}}$, with detailed results provided in the Appendix. We evaluate performance using two widely adopted metrics: Area Under the Receiver Operating Characteristic Curve (*AUC-ROC*), which measures the model's ability to distinguish between normal and anomalous nodes, and Area Under the Precision-Recall Curve (*AUC-PR*), which is particularly informative in imbalanced scenarios by emphasizing precision on the minority (anomalous) class.

## 4.2 EMPIRICAL RESULTS

**Small-scale GAD Performance.** We first perform the performance comparison of **DEMO** on three small-scale graph datasets, the results are shown in Table 1. As shown, **DEMO** consistently outperforms all existing baselines across both AUC-ROC and AUC-PR metrics. ❶ *Among unsupervised methods,* **DEMO** *achieves substantial improvements over all competitors.* For instance, on the Photo dataset, it surpasses the best-performing unsupervised baseline (CoLA) by 60.61% in AUC-ROC, and the gain in AUC-PR is even more pronounced. Across three datasets, **DEMO** achieves 80.99% higher average AUC-ROC (*vs.* 0.4956 of CoLA) and 368.11% improvement in average AUC-PR (*vs.* GAAN's 0.1539). These results indicate that unsupervised approaches relying solely on structural reconstruction or embedding learning are limited in capturing the diversity of anomaly types under open-set conditions, resulting in poor generalization. ❷ *In the semi-supervised setting,* **DEMO** *maintains dominant advantages, notably outperforming strong baselines such as* NSReg. On average, across the three datasets, **DEMO** achieves a 7.86% improvement in AUC-ROC over the second-best method (NSReg), and a 17.60% gain in AUC-PR compared to GGAD. This highlights that beyond effectively leveraging limited labeled data, **DEMO** benefits from its diverse anomaly modeling and adaptive mechanisms, yielding stronger recognition of unseen anomalies. ❸ In summary, the comprehensive performance of **DEMO** on small-scale datasets validates its effectiveness in the open-set GAD task, particularly under conditions of sparse anomaly distribution and limited supervision, where unified and robust modeling of both seen and unseen anomalies is essential.

**Large-scale GAD Performance.** Furthermore, we present the performance evaluation results of **DEMO** on three large-scale graph datasets in Table 2. ❶ *As observed,* **DEMO** *demonstrates strong robustness and broad adaptability in large-scale open-set scenarios.* On the Yelp dataset, **DEMO** achieves an AUC-ROC of 0.7097, outperforming the second-best method NSReg by 1.16%. Although its AUC-PR is slightly lower than NSReg, it still exhibits reliable and stable performance. On the two extremely large datasets, **DEMO** delivers consistently strong results, improving AUC-ROC by 2.33% and 1.17%, and boosting AUC-PR by 0.85% and 25.58%, respectively, compared to the strongest baselines. ❷ These results confirm that **DEMO** maintains reliable performance even on large-scale graphs, and is particularly effective in handling complex topologies and extreme class imbalance by robustly expanding decision boundaries and accurately identifying previously unseen anomalies.

**Data Efficiency.** To further evaluate the impact of anomaly sample quantity on model performance, we conducted experiments by varying the number of anomalous nodes in the training set across three levels: 20, 50, and 100. These experiments were performed on three datasets. Additional results and comparisons are provided in the Appendix. Figure 2 (a-c) illustrates how each method's AUC-ROC score changes with different amounts of anomaly supervision. As shown in the figure,

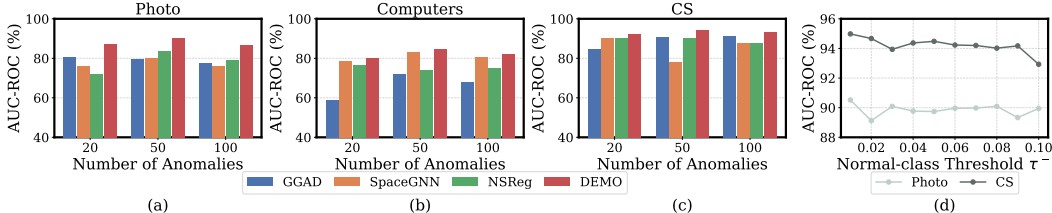

Figure 2: (a), (b) and (c) Comparison of model performance under different numbers of training anomalies across three datasets. (d) Sensitivity analysis of the threshold $\tau^-$.

**DEMO** consistently maintains the best or near-best performance across all datasets as the number of anomalies increases, and notably demonstrates clear advantages in low-resource settings, such as when 20 anomalous nodes are available. On the CS dataset, even with just 20 anomaly nodes, **DEMO** achieves an AUC-ROC of approximately 0.90, showing a substantial gap. On the Computers dataset, **DEMO** reaches its peak performance with 50 anomalies, significantly outperforming all baselines. In summary, **DEMO** exhibits strong anomaly detection capabilities under severe data scarcity and maintains excellent scalability and training efficiency as the number of anomaly samples grows, confirming its advantage in efficient use of limited supervision.

**Sensitivity Analysis.** In this part, we conduct a sensitivity analysis on the hyperparameters $\tau^+$ and $\tau^-$. Figure 2 (d) illustrates the impact of the predefined normal-class threshold $\tau^-$ on the detection performance under our proposed class-aware threshold adjustment strategy. Since the anomaly-class threshold is defined as $1 - \tau^-$, this experiment effectively evaluates the joint influence of both class thresholds. We vary $\tau^-$ within the range of $[0.01, 0.1]$ and examine its effect on AUC-ROC across two datasets. The results show that both datasets achieve optimal performance when $\tau^- = 0.01$, indicating a well-balanced separation between normal and anomalous classes. As $\tau^-$ decreases (*i.e.*, the anomaly threshold increases), the model performance begins to fluctuate or decline, suggesting that an overly relaxed anomaly selection criterion may introduce low-confidence samples into training and compromise learning. This demonstrates that setting a relatively lower normal-class threshold (or, equivalently, a higher anomaly-class threshold) enables the model to more reliably select high-confidence pseudo-labeled samples, effectively mitigating label bias and improving overall detection performance. Therefore, we set $\tau^+$ and $\tau^-$ to 0.99 and 0.01, respectively.

**Ablation Study.** We introduce several variants of **DEMO** to analyze the contribution of each component: (1) **DEMO** w/o All removes all three components; (2) **DEMO** w/o Mix removes the multi-sample fusion component; (3) **DEMO** w/o EG removes the energy gradient-guided optimization strategy; and (4) **DEMO** w/o PL disables the pseudo-labeling component. As shown in Table 3, the ablation results reveal that each component plays a vital role in enhancing model performance. First, removing the PL component results in the most substantial performance drop across both datasets, demonstrating the crucial impact of class-aware pseudo-labeling in mitigating label scarcity and improving anomaly detection. Second, the exclusion of the EG module consistently reduces performance, validating the effectiveness of gradient-based reweighting in prioritizing informa-

Table 3: Ablation Study on two benchmarks.

| Datasets | Photo | | Computers | |
|---|---|---|---|---|
| Metrics | AR | AP | AR | AP |
| **DEMO** w/o All | 0.8300 | 0.5692 | 0.7576 | 0.5325 |
| **DEMO** w/o Mix | 0.8750 | 0.6023 | 0.8197 | 0.6292 |
| **DEMO** w/o EG | 0.8849 | 0.6171 | 0.8100 | 0.5998 |
| **DEMO** w/o PL | 0.8616 | 0.6150 | 0.8094 | 0.5949 |
| **DEMO** | **0.9023** | **0.6330** | **0.8439** | **0.6458** |

tive and uncertain samples during training. Third, the absence of the Mix module also causes noticeable degradation, highlighting the benefit of synthesizing diverse anomaly representations to better expand the decision boundary. Additionally, the variant **DEMO** w/o All, which disables all three components, yields the lowest overall performance, indicating that the full combination of modules is essential for achieving the best anomaly detection results. Overall, these results confirm that all three modules contribute synergistically to the robustness and generalization of **DEMO**.

**Visualization Results.** To further assess the representation quality of node embeddings learned by different models, we visualize the 2D t-SNE projections of node features on the Photo and Computers datasets, as shown in Figure 3. Blue and green points correspond to anomaly and normal nodes, respectively. On both datasets, **DEMO** produces clearly separated clusters between anomalous and normal nodes. The anomalies (in blue) are more tightly grouped and distinctly detached from the

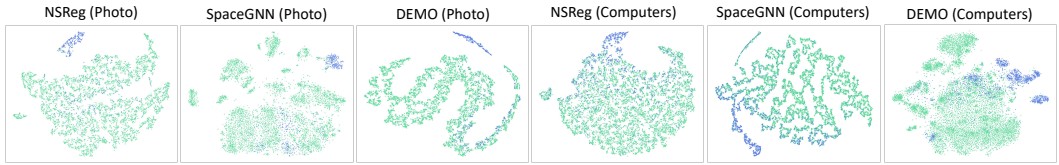

Figure 3: Visualization results of the features of the three algorithms on the two datasets.

normal class, indicating that **DEMO** effectively enhances anomaly-specific feature encoding and decision boundary clarity. In contrast, the embeddings generated by NSReg and SpaceGNN exhibit substantial overlap between the two classes, particularly on the Computers dataset, where anomalies appear largely entangled with normal nodes.

## 5 CONCLUSION

This paper presents **DEMO**, a novel framework for open-set graph anomaly detection. By combining multi-sample fusion with gradient-guided weight adjustment, **DEMO** enhances model generalization to unseen anomalies. To mitigate the effects of label scarcity, it further incorporates a class-aware threshold adaptation scheme guided by historical records for reliable pseudo-label generation. Extensive experiments validate its effectiveness across diverse datasets. In addition to strong empirical performance, **DEMO** demonstrates robustness across varying data scales and supervision levels, making it broadly applicable to real-world graph anomaly detection tasks.

### ACKNOWLEDGMENTS

This work is supported in part by the National Natural Science Foundation of China under Grants 62306014, the Postdoctoral Fellowship Program (Grade A) of CPSF under Grant BX20250376, the Sichuan Science and Technology Program under Grant 2025ZNSFSC1506, the Fundamental Research Funds for the Central Universities under Grant 1082204112K97, and the Sichuan University Interdisciplinary Innovation Fund.

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

## A STATEMENT ON THE USE OF LARGE LANGUAGE MODELS

During the preparation of this manuscript, we used the Large Language Model (LLM) to polish the language and correct grammatical errors to improve readability. The LLM was not involved in any core research aspects of the paper, such as research ideation, experimental design, or analysis of results.

## B ALGORITHM

The complete training algorithm of our proposed **DEMO** framework is outlined in Algorithm 1. We divide the training process into three successive parts. Each part corresponds to a core component of **DEMO**, including mixup anomalies generation, adaptive weighting, and pseudo-label generation.

---

**Algorithm 1** Training Procedure of the Proposed **DEMO** Framework

---

**Require:** Graph $\mathcal{G} = (\mathcal{V}, \mathcal{E}, \mathbf{X})$, training set $\mathcal{V}^{\text{train}}$, training anomalies $\mathcal{V}_a^{\text{train}}$, unlabeled set $\mathcal{V}_u$, training epochs $T$
**Ensure:** Trained model parameters $\theta$
1: Initialize model parameters $\theta$
2: Initialize memory bank $\mathcal{M} \leftarrow \emptyset$
3: **for** $t$ in $1, 2, \ldots, T$ **do**
4:      Part I: Multi-sample Mixup for Unseen Anomaly Simulation
5:      **for all** anomaly node $v_i^{\text{train}} \in \mathcal{V}_a^{\text{train}}$ **do**
6:          Generate mixup embeddings $\hat{z}_i = \sum_j \alpha_{ij} z_j$ using similarity-based weights $\alpha_{ij}$ (Eq. 1)
7:          Obtain augmented views $\hat{z}_i^a, \hat{z}_i^b$ via node augmentation
8:          Calculate consistency loss $\mathcal{L}_{\text{cons}}$ between views
9:      **end for**
10:     Part II: Energy Gradient-Based Weight Adaptation
11:     **for all** training node $v_i^{\text{train}} \in \mathcal{V}^{\text{train}}$ **do**
12:         Compute energy $E_\theta(v_i)$ and influence score $T_{\text{val}}(v_i)$
13:         Derive adaptive weight $\beta_{v_i}$ based on energy gradient (Eq. 7)
14:     **end for**
15:     Part III: Pseudo-Labeling Generation
16:     Update memory bank $\mathcal{M}$ with selection history from epoch $t$
17:     **for all** unlabeled node $v_j \in \mathcal{V}_u$ **do**
18:         Predict probability $\hat{p}_t(v_j)$ for each class
19:         Compute class-specific adaptive thresholds $\tau_t^+$ and $\tau_t^-$ (Eq. 10)
20:         **if** $\hat{p}_t(v_j) \geq \tau_t^+$ or $\hat{p}_t(v_j) \leq \tau_t^-$ **then**
21:            Assign pseudo-label $\hat{y}_j$ to $v_j$ based on threshold rule
22:            Add $(v_j, \hat{y}_j)$ to $\mathcal{V}_l$
23:         **end if**
24:     **end for**
25:     Compute total loss in Eq. 11
26:     Update $\theta$ using optimizer
27: **end for**
28: **return** Final model parameters $\theta$

---

## C PROOF OF THEOREM

Since more similar samples are inherently more challenging for the model to distinguish, leveraging such samples can enhance the robustness of the model. Therefore, we desire that the synthesized samples generated through the mixup operation retain similarity to the original seen anomaly samples, yet differ sufficiently to approximate the distribution of unseen anomalies. Specifically, by using similarity as weights in the mixup process, we aim to achieve:

$$\mathcal{S}\left(\hat{z}_i, z_j\right) \approx \mathcal{S}\left(z_i, z_j\right), \tag{12}$$

where $\mathcal{S}$ denotes the inner-product similarity function. Thus, the synthesized samples maintain similar levels of similarity to original samples after mixup. Note that both $z_i$ and $z_j$ are anomaly training samples ($z_i^{\text{train}}$ and $z_j^{\text{train}}$), abbreviated here for clarity.

**Proof of Theorem 3.1**: The similarity between the synthesized sample $\hat{z}_i$ and an original sample $z_j$ can be expressed as:

$$\mathcal{S}\left(\hat{z}_i, z_j\right) = \left(\sum_{k=1}^{N} \alpha_{ik} z_k\right)^{\top} z_j = \sum_{k=1}^{N} \alpha_{ik} \mathcal{S}\left(z_k, z_j\right). \tag{13}$$

We separate the contribution from $z_j$ and other samples:

$$\mathcal{S}\left(\hat{z}_i, z_j\right) = \alpha_{ij} \mathcal{S}\left(z_j, z_j\right) + \sum_{k \neq j} \alpha_{ik} \mathcal{S}\left(z_k, z_j\right) \tag{14}$$

Since the weights $\alpha_{ik}$ are obtained via a softmax function, it follows that if $\mathcal{S}\left(z_i, z_j\right) \geq \mathcal{S}\left(z_i, z_k\right)$, then $\alpha_{ij} \geq \alpha_{ik}$. Thus, the weights concentrate on highly similar samples. Assuming the similarity between $z_i$ and $z_j$ is the highest among all other samples, it follows that $\alpha_{ij}$ significantly exceeds the remaining weights.

Further, observing that $\mathcal{S}\left(z_j, z_j\right) = \|z_j\|^2$, and letting $\mathcal{S}\left(z_k, z_j\right) = \mathcal{S}\left(z_i, z_j\right) + \delta_{kj}$, where $\delta_{kj} = \mathcal{S}\left(z_k, z_j\right) - \mathcal{S}\left(z_i, z_j\right)$, we substitute into the above equation:

$$\mathcal{S}\left(\hat{z}_i, z_j\right) = \alpha_{ij} \|z_j\|^2 + \sum_{k \neq j} \alpha_{ik} \left[\mathcal{S}\left(z_i, z_j\right) + \delta_{kj}\right]. \tag{15}$$

Rearranging terms, we have:

$$\mathcal{S}\left(\hat{z}_i, z_j\right) = \mathcal{S}\left(z_i, z_j\right) \cdot \sum_{k \neq j} \alpha_{ik} + \alpha_{ij} \|z_j\|^2 + \sum_{k \neq j} \alpha_{ik} \delta_{kj} \tag{16}$$

Given that $\sum_{k=1}^{N} \alpha_{ik} = 1$, this simplifies to:

$$\mathcal{S}\left(\hat{z}_i, z_j\right) = \mathcal{S}\left(z_i, z_j\right) \cdot \left(1 - \alpha_{ij}\right) + \alpha_{ij} \|z_j\|^2 + \sum_{k \neq j} \alpha_{ik} \delta_{kj}. \tag{17}$$

Since $\delta_{kj}$ can be positive or negative, we cannot directly determine the sign of the summation $\sum_{k \neq j} \alpha_{ik} \delta_{kj}$. However, we can bound its magnitude by applying the triangle inequality:

$$\sum_{k \neq j} \alpha_{ik} \delta_{kj} \geq -\sum_{k \neq j} \alpha_{ik} \left|\delta_{kj}\right|. \tag{18}$$

Thus, we have:

$$\mathcal{S}\left(\hat{z}_i, z_j\right) \geq \mathcal{S}\left(z_i, z_j\right) \cdot \left(1 - \alpha_{ij}\right) + \alpha_{ij} \|z_j\|^2 - \sum_{k \neq j} \alpha_{ik} \left|\delta_{kj}\right|. \tag{19}$$

Letting $\epsilon = \sum_{k \neq j} \alpha_{ik} \left|\delta_{kj}\right|$, we rewrite this as:

$$\mathcal{S}\left(\hat{z}_i, z_j\right) \geq \mathcal{S}\left(z_i, z_j\right) \cdot \left(1 - \alpha_{ij}\right) + \alpha_{ij} \|z_j\|^2 - \epsilon. \tag{20}$$

Despite $z_j$ being highly similar to $z_i$, it is evident that the similarity between identical samples is greater than between different samples, thus, $\alpha_{ij} \|z_j\|^2 \geq \alpha_{ij} \mathcal{S}\left(z_i, z_j\right)$. Consequently, we have:

$$\mathcal{S}\left(\hat{z}_i, z_j\right) \geq \mathcal{S}\left(z_i, z_j\right) \cdot \left(1 - \alpha_{ij}\right) + \alpha_{ij} \mathcal{S}\left(z_i, z_j\right) - \epsilon = \mathcal{S}\left(z_i, z_j\right) - \epsilon. \tag{21}$$

**Interpretation of $\epsilon$:** The value $\epsilon$ quantifies the cumulative disturbance caused by non-target samples ($k \neq j$) in the synthesized sample's similarity. It is influenced by two factors: 1) Concentration of the weights $\alpha_{ik}$ (greater concentration implies smaller $\epsilon$). 2) Similarity differences $\left|\delta_{kj}\right|$ among samples (smaller differences imply smaller $\epsilon$).

Table 4: Statistics of 6 datasets including the number of nodes, edges, and dimensions, the number of classes, the number of anomalous classes, the number of anomalous nodes, and the selection interval for anomalous nodes.

| Dataset | #Nodes | #Edge | #Dimension | #Classes | #Ano. Classes | # Ano. | #Ano. Prop |
|---|---|---|---|---|---|---|---|
| Photo | 7,650 | 238,162 | 745 | 8 | 2 | 700 | 0%-5% |
| Computers | 13,752 | 491,722 | 767 | 10 | 5 | 2,064 | 0%-5% |
| CS | 18,333 | 163,788 | 6,805 | 15 | 8 | 4,159 | 0%-5% |
| Yelp | 45,954 | 3,846,979 | 32 | 3 | 2 | 6,677 | 0%-5% |
| ogbn-arxiv | 169,343 | 1,166,243 | 128 | 40 | 4 | 27,830 | 3%-5% |
| ogbn-mag | 736,389 | 5,416,271 | 128 | 349 | 15 | 3,135 | 0%-0.03% |

## D  DATASETS DETAILS

We evaluate the performance of **DEMO** on open-set graph anomaly detection using six real-world graph datasets. Notably, there are currently no dedicated benchmark datasets tailored for open-set GAD tasks. Therefore, we adapt existing datasets through customized processing to simulate the open-set scenario. Specifically, we conduct experiments on three small-scale datasets (Photo, Computers, and CS) and three large-scale datasets (Yelp, ogbn-arxiv, and ogbn-mag). Table 4 summarizes the dataset processing details, where "Ano. Classes" denotes the number of anomalous classes, "Ano" indicates the total number of anomalous nodes, and "Ano. Prop" reflects the proportion range of node count in each class used to determine whether it is selected as an anomaly class. To emulate the open-set setting, we designate classes with low sample counts as anomaly classes, creating a mismatch between the class distribution of the training and testing phases. This better reflects real-world scenarios where unknown or unseen anomaly types may emerge at test time, posing a significant challenge for generalization. Additional descriptions of each dataset are provided below.

- **Photo and Computers** (McAuley et al., 2015). Amazon Computers and Amazon Photo are subsets of the Amazon co-purchase graph, where nodes represent products, and edges indicate co-purchase relationships. The node features are encoded as bag-of-words from product reviews, and class labels correspond to the product category.

- **CS.** The Computer Science (CS) section of the Coauthor dataset is used for node classification tasks. In this dataset, nodes represent authors connected by an edge if they have co-authored a paper. The node features capture the keywords of the authors' papers, while the class labels represent the most active research fields of each author.

- **Yelp** (Zeng et al., 2019).The task of Yelp dataset is categorizing types of businesses based on customer reviewers and friendship. The Yelp dataset is a heterogeneous graph with three distinct views. For our experiments, we focus on the "Review-User-Review (RUR)" edge subset.

- **ogbn-arxiv** (Mikolov et al., 2013). The ogbn-arxiv dataset is a citation network of Computer Science (CS) arXiv papers, where each node represents a paper and each directed edge indicates a citation. The task is to predict the 40 subject areas of these papers, using a 40-class classification approach, with a realistic data split based on publication dates for training, validation, and testing.

- **ogbn-mag** (Wang et al., 2020). The ogbn-mag dataset is a heterogeneous network derived from the Microsoft Academic Graph (MAG), consisting of four types of entities: papers, authors, institutions, and fields of study. The task is to predict the venue (conference or journal) of each paper, given its content, references, authors, and affiliations, which is formulated as a 349-class classification problem. For our experiments, we focus on the "paper" nodes, using the ("paper", "citeps", "paper") edges to represent citations between papers.

## E  BASELINES DETAILS

In our experiments, we compare **DEMO** with 12 representative GAD methods, which are categorized into two groups: unsupervised and semi-supervised approaches. Notably, TAM, OCGNN, ANOMA-LOUS, and DOMINANT are originally unsupervised models; to incorporate supervision for fair comparison, we adopt the semi-supervised design strategy proposed in GGAD. Specifically, for

TAM, we refine its affinity maximization objective to focus exclusively on labeled normal nodes. In OCGNN, the one-class center optimization is constrained to labeled normal instances. For DOMINANT and AnomalyDAE, we restrict their auto-encoding loss computation to labeled normal nodes during training. Below, we provide brief descriptions of these 12 methods.

- **ANOMALOUS** (Peng et al., 2018). ANOMALOUS proposes a joint framework for anomaly detection on attributed networks that integrates attribute selection and anomaly detection using CUR decomposition and residual analysis. It filters out noisy and irrelevant node attributes, improving detection performance by focusing on the most representative attributes.

- **DOMINANT** (Ding et al., 2019). DOMINANT proposes a deep learning model that combines Graph Convolutional Networks (GCN) and autoencoders for anomaly detection in attributed networks, addressing challenges like sparsity and nonlinearity by measuring reconstruction errors from both structural and attribute perspectives.

- **AnomalyDAE** (Fan et al., 2020). AnomalyDAE incorporates an attention mechanism to capture structure patterns and uses both node and attribute embeddings to model cross-modality interactions during reconstruction, enabling effective anomaly detection.

- **GAAN** (Chen et al., 2020b). GAAN generates fake graph nodes using Gaussian noise and employs an encoder to map both real and fake nodes into a latent space, using a discriminator to distinguish between real and fake nodes. Anomaly detection is then performed by evaluating reconstruction errors and identification confidence.

- **CoLA** (Liu et al., 2021). CoLA uses a contrastive self-supervised learning framework for anomaly detection in attributed networks, which captures local information by sampling contrastive instance pairs and utilizing a GNN-based contrastive learning model. The framework targets anomaly detection through a specific learning objective and adapts to large networks by training on batches of instance pairs rather than the full graph.

- **CONAD** (Xu et al., 2022). CONAD introduces a framework that integrates human knowledge of different anomaly types into attributed network anomaly detection. It employs a novel data augmentation strategy to model prior human knowledge, which is then incorporated into a Siamese graph neural network encoder with contrastive loss. Anomalies are detected by ranking nodes based on their reconstruction error.

- **ConsisGAD** (Chen et al., 2024). ConsisGAD is a model designed for graph anomaly detection in settings with limited supervision and class imbalance. It leverages unlabeled data through consistency training and a learnable data augmentation mechanism, while utilizing homophily distribution variance to improve class distinction using a simplified GNN backbone.

- **GGAD** (Qiao et al., 2024). GGAD introduces a semi-supervised generative approach for graph anomaly detection, aiming to better utilize known normal nodes by generating pseudo anomaly nodes ("outlier nodes") for training a one-class classifier. It leverages priors on asymmetric local affinity and egocentric closeness to generate reliable outlier nodes, thus enhancing detection performance in the absence of ground truth for real anomalies.

- **TAM** (Qiao & Pang, 2023). TAM introduces a novel unsupervised anomaly scoring measure based on local node affinity, where normal nodes exhibit stronger connections with each other than abnormal nodes. It employs Truncated Affinity Maximization (TAM) to learn node representations that maximize local affinity, optimizing on truncated graphs to remove non-homophily edges and improve anomaly detection performance.

- **OCGNN** (Wang et al., 2021). OCGNN introduces a one-class classification framework for graph anomaly detection, combining the powerful representation ability of graph neural networks with the classical one-class support vector machine objective. This approach addresses the limitations of traditional anomaly detection methods in graph data, achieving significant improvements over existing baselines.

- **SpaceGNN** (Dong et al., 2025a) SpaceGNN introduces a model for node anomaly detection with limited labels, using a Learnable Space Projection function and a Distance Aware Propagation module to enhance node representations and information propagation. It outperforms data augmentation techniques and achieves better results than existing methods.

- **NSReg** (Wang et al., 2023b). NSReg introduces a novel approach for open-set graph anomaly detection by adding a regularization term to enforce compact, semantically-rich representations of

Table 5: AUC-ROC and AUC-PR on the unseen anomaly classes on three small-scale datasets. The best performance is boldfaced, with the second-best underlined.

| Datasets | Photo | | Computers | | CS | |
|---|---|---|---|---|---|---|
| Metrics | AUC-ROC | AUC-PR | AUC-ROC | AUC-PR | AUC-ROC | AUC-PR |
| Anomalous | $0.4841_{\pm0.015}$ | $0.0435_{\pm0.010}$ | $0.5640_{\pm0.075}$ | $0.1435_{\pm0.008}$ | $0.2364_{\pm0.018}$ | $0.1323_{\pm0.004}$ |
| DOMINANT | $0.4721_{\pm0.021}$ | $0.0548_{\pm0.004}$ | $0.5597_{\pm0.078}$ | $0.1474_{\pm0.003}$ | $0.3836_{\pm0.026}$ | $0.1601_{\pm0.003}$ |
| AnomalyDAE | $0.4798_{\pm0.019}$ | $0.0495_{\pm0.007}$ | $0.5793_{\pm0.045}$ | $0.1530_{\pm0.006}$ | $0.3724_{\pm0.019}$ | $0.1566_{\pm0.006}$ |
| GAAN | $0.4708_{\pm0.031}$ | $0.0432_{\pm0.002}$ | $0.5739_{\pm0.020}$ | $0.1620_{\pm0.012}$ | $0.4636_{\pm0.015}$ | $0.1845_{\pm0.009}$ |
| CoLA | $0.7064_{\pm0.024}$ | $0.0808_{\pm0.006}$ | $0.4565_{\pm0.018}$ | $0.1149_{\pm0.010}$ | $0.4411_{\pm0.029}$ | $0.1792_{\pm0.005}$ |
| CONAD | $0.4746_{\pm0.024}$ | $0.0564_{\pm0.011}$ | $0.5560_{\pm0.027}$ | $0.1436_{\pm0.011}$ | $0.3838_{\pm0.035}$ | $0.1602_{\pm0.010}$ |
| ConsisGAD | $0.5571_{\pm0.059}$ | $0.0563_{\pm0.012}$ | $0.5057_{\pm0.026}$ | $0.1325_{\pm0.019}$ | $0.7364_{\pm0.026}$ | $0.4594_{\pm0.018}$ |
| GGAD | $0.6274_{\pm0.043}$ | $0.0679_{\pm0.004}$ | $0.6636_{\pm0.097}$ | $0.2471_{\pm0.054}$ | $\underline{0.8931}_{\pm0.059}$ | $\underline{0.7596}_{\pm0.039}$ |
| TAM | $0.5979_{\pm0.027}$ | $0.0604_{\pm0.003}$ | $0.4458_{\pm0.049}$ | $0.1159_{\pm0.007}$ | $0.6692_{\pm0.018}$ | $0.3396_{\pm0.023}$ |
| OCGNN | $0.5208_{\pm0.015}$ | $0.0563_{\pm0.004}$ | $0.5112_{\pm0.013}$ | $0.1308_{\pm0.005}$ | $0.7662_{\pm0.056}$ | $0.4406_{\pm0.027}$ |
| ANO-S | $0.5315_{\pm0.032}$ | $0.0573_{\pm0.007}$ | $0.4508_{\pm0.061}$ | $0.1168_{\pm0.002}$ | $0.8181_{\pm0.082}$ | $0.5727_{\pm0.052}$ |
| DOM-S | $0.5705_{\pm0.003}$ | $0.0589_{\pm0.001}$ | $0.4424_{\pm0.046}$ | $0.1118_{\pm0.008}$ | $0.8239_{\pm0.058}$ | $0.5645_{\pm0.033}$ |
| SpaceGNN | $0.6808_{\pm0.031}$ | $0.0733_{\pm0.007}$ | $\underline{0.8108}_{\pm0.042}$ | $\underline{0.5010}_{\pm0.025}$ | $0.8145_{\pm0.065}$ | $0.6337_{\pm0.041}$ |
| NSReg | $\underline{0.7369}_{\pm0.014}$ | $\underline{0.0956}_{\pm0.006}$ | $0.6846_{\pm0.013}$ | $0.3547_{\pm0.036}$ | $0.8863_{\pm0.079}$ | $0.7460_{\pm0.027}$ |
| **DEMO** | $\mathbf{0.8202}_{\pm0.019}$ | $\mathbf{0.1303}_{\pm0.003}$ | $\mathbf{0.8119}_{\pm0.036}$ | $\mathbf{0.5032}_{\pm0.015}$ | $\mathbf{0.9354}_{\pm0.039}$ | $\mathbf{0.8417}_{\pm0.017}$ |

normal nodes. This helps the model generalize to unseen anomalies while reducing false negatives and improving detection performance.

# F  IMPLEMENTATION DETAILS

## F.1  EXPERIMENTAL PROTOCOL

In our open-set graph anomaly detection (GAD) experiments, we follow a process that alternates between treating different anomaly classes as "seen" anomalies and considering the remaining anomaly classes as "unseen" anomalies. For each dataset, we first select a class to be treated as seen and train the model using this class along with normal nodes. The training set consists of the normal nodes and the selected anomaly class. Next, we evaluate the model's performance by measuring AUC-ROC and AUC-PR scores on all anomaly classes (results in Section 4.2) and only on the unseen anomaly classes (results can be found in Section G). Additionally, for smaller datasets (Photo, Computers, CS), we repeat this process five times and record the average and variance. For larger datasets (Yelp, ogbn-arxiv, ogbn-mag), we conduct a single experiment due to the large volume of data. We use the same data split for training the baselines to ensure fairness.

## F.2  HYPERPARAMETER SETTING

In addition to the implementation details described in Section 4.1, we made the following hyperparameter settings. According to (Wang et al., 2023b), we set the number of neighbors for GraphSAGE aggregation to 25 for the first layer and 10 for the second layer, except for the ogbn-mag dataset, to improve computational efficiency. We configured the GraphSAGE model with 2 layers, set the dropout rate to 0.5, and the batch size to 512. For the reproduction of unsupervised methods, we referred to the code and would like to thank the authors for making their implementation publicly available.

# G  MORE EXPERIMENTS

## G.1  GAD PERFORMANCE EXTENSION EXPERIMENT

We evaluate the detection capabilities of all methods on unseen anomaly classes. Specifically, Tables 5 and 6 present the performance of each algorithm on three small-scale datasets (Photo, Computers, and CS) and three large-scale datasets (Yelp, ogbn-arxiv, and ogbn-mag), respectively, where seen anomaly

Table 6: AUC-ROC and AUC-PR on the unseen anomaly classes on three large-scale datasets. The best performance is boldfaced, with the second-best underlined.

| Datasets | Metrics | ConsisGAD | GGAD | TAM | OCGNN | ANO-S | DOM-S | SpaceGNN | NSReg | **DEMO** |
|---|---|---|---|---|---|---|---|---|---|---|
| Yelp | AUC-ROC | 0.5238 | 0.6819 | 0.6016 | 0.7185 | 0.7168 | 0.7056 | 0.5212 | 0.4941 | **0.7235** |
| | AUC-PR | 0.0205 | **0.0776** | 0.0351 | 0.0625 | 0.0538 | 0.0381 | 0.0220 | 0.0178 | 0.0635 |
| ogbn-arxiv | AUC-ROC | 0.5254 | 0.5101 | OOM | OOM | 0.4462 | 0.4474 | 0.5180 | 0.5356 | **0.5643** |
| | AUC-PR | 0.1655 | 0.1473 | OOM | OOM | 0.1187 | 0.1204 | 0.1611 | 0.1566 | **0.1808** |
| ogbn-mag | AUC-ROC | 0.4879 | OOM | OOM | OOM | OOM | OOM | 0.4425 | 0.4628 | **0.4995** |
| | AUC-PR | 0.0039 | OOM | OOM | OOM | OOM | OOM | 0.0036 | 0.0036 | **0.0053** |

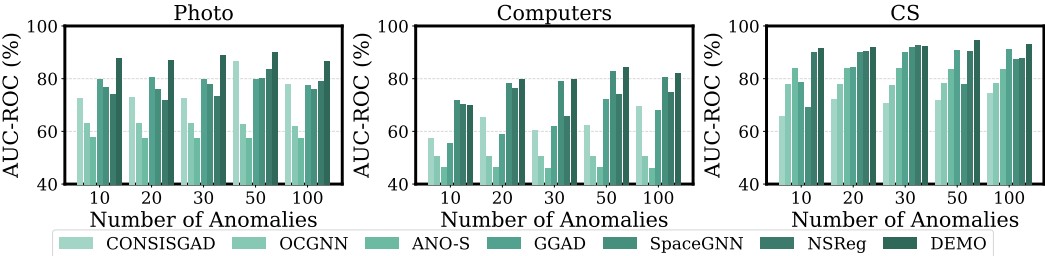

Figure 4: The AUC-ROC under different numbers of training anomalies across three datasets.

nodes are excluded from the test set. Only normal and unseen anomaly nodes are considered in the computation of AUC-ROC and AUC-PR. Overall, **DEMO** consistently achieves the best performance across both categories of datasets, further confirming its strong generalization ability to unknown anomalies in realistic open-set GAD scenarios. Across all six datasets, **DEMO** not only outperforms all unsupervised baselines but also significantly exceeds other competitive semi-supervised methods, especially in AUC-PR. Specifically, **DEMO** achieves the highest AUC-ROC and AUC-PR scores on all three small-scale datasets. For example, on the Photo dataset, **DEMO** reaches an AUC-ROC of 0.8202, notably surpassing the second-best method NSReg (0.7369), while also achieving an AUC-PR of 0.1303, substantially outperforming all baselines. Similar trends are observed on Computers and CS, indicating that **DEMO** effectively detects diverse and distribution-shifted unseen anomalies. On large-scale datasets, **DEMO** again obtains the best and near-best results on Yelp and ogbn-arxiv, with AUC-ROC scores of 0.7235 and 0.5643, showing significant improvements over all other methods. Even on the most challenging ogbn-mag dataset, characterized by extreme class imbalance and a large graph size, **DEMO** maintains its advantage, achieving an AUC-ROC of 0.4995 and an AUC-PR of 0.0053, demonstrating high sensitivity to low-frequency unseen anomalies. In summary, **DEMO** exhibits strong robustness and generalizability even when seen anomalies are excluded from evaluation, validating its effectiveness in identifying unknown patterns in open-set graph anomaly detection.

## G.2 DATA EFFICIENCY EXTENSION EXPERIMENT

We investigate the impact of the number of seen anomalous nodes in the training set on model performance. Specifically, we conduct experiments on three small-scale datasets (Photo, Computers, and CS) by varying the number of training anomalies across five levels: 10, 20, 30, 50, and 100. We evaluate the performance of all methods using AUC-ROC (Figure 4) and AUC-PR (Figure 5) to analyze their adaptability and robustness under both low-resource and high-resource settings. Overall, **DEMO** consistently outperforms all baselines across different settings, particularly exhibiting a clear advantage under low-resource scenarios (*e.g.*, with only 10 or 20 anomalies), indicating its strong training efficiency and generalization ability in anomaly-scarce situations. More specifically, on the Photo dataset, **DEMO** achieves the best results on both AUC-ROC and AUC-PR in most configurations, with stable performance especially at 20 and 30 anomalies. On the Computers dataset, performance improves steadily as more anomalies are added. On the CS dataset, **DEMO** demonstrates high overall stability, maintaining the best performance even with as few as 10 anomalies, and consistently achieving top scores across different anomaly scales. In conclusion, **DEMO** demonstrates stable and

Table 7: Loss weights of six datasets

| Datasets | $\lambda_{un}$ | $\lambda_{mix}$ | $\lambda_{eng}$ |
|---|---|---|---|
| Photo | 0.5 | 0.1 | 0.1 |
| Computers | 0.1 | 0.1 | 0.5 |
| CS | 0.3 | 0.1 | 0.3 |
| Yelp | 0.3 | 0.1 | 0.1 |
| ogbn-arxiv | 0.5 | 0.1 | 0.1 |
| ogbn-mag | 0.5 | 0.1 | 0.1 |

Table 8: Ablation Study on the other three benchmarks.

| Datasets | CS | | Yelp | | ogbn-arxiv | |
|---|---|---|---|---|---|---|
| Metrics | AR | AP | AR | AP | AR | AP |
| w/o All | 0.7815 | 0.6394 | 0.5942 | 0.1063 | 0.4219 | 0.1944 |
| w/o Mix | 0.8841 | 0.8151 | 0.6519 | 0.1927 | 0.5061 | 0.2455 |
| w/o EG | 0.9027 | 0.7964 | 0.6637 | 0.1833 | 0.5337 | 0.2849 |
| w/o PL | 0.8364 | 0.8005 | 0.6362 | 0.1624 | 0.4868 | 0.2517 |
| DEMO | **0.9448** | **0.8857** | **0.7097** | **0.2238** | **0.6364** | **0.3329** |

superior detection capability across different levels of anomaly supervision, with its exceptional performance under limited anomaly samples further validating its data efficiency and robustness in open-set GAD.

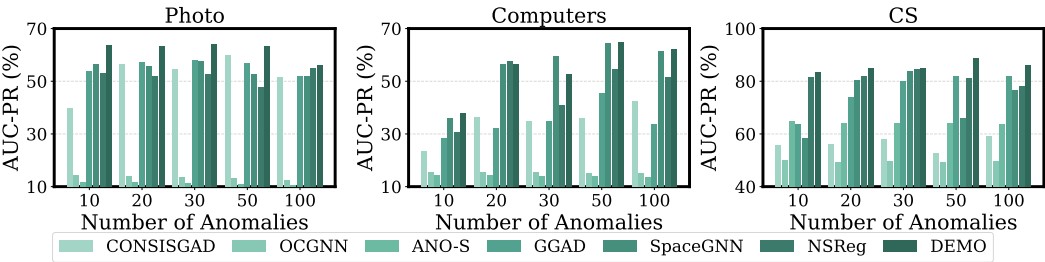

Figure 5: The AUC-PR under different numbers of training anomalies across three datasets.

### G.3 MORE ABLATION STUDY

To further validate our method, we present the ablation study results on three additional benchmarks in Table 8. The results consistently corroborate the conclusions from our main analysis. First, removing the PL component results in a substantial performance drop across all three datasets, particularly on CS and ogbn-arxiv, demonstrating the critical impact of our class-aware pseudo-labeling in complex scenarios. Second, the exclusion of the EG and Mix modules also leads to a clear degradation in performance, validating their respective effectiveness in prioritizing informative samples and synthesizing diverse anomaly representations. Additionally, the variant DEMO w/o All consistently yields the lowest performance, confirming that the full combination of modules is essential for achieving optimal results. Overall, these extensive results provide further evidence that all three components contribute synergistically to the robustness and superior performance of DEMO.

### G.4 PARAMETERS SENSITIVITY

We conduct a sensitivity analysis of the loss weights on two datasets (Photo and Computers) to evaluate the contribution and robustness of each component to both overall performance and unseen anomaly

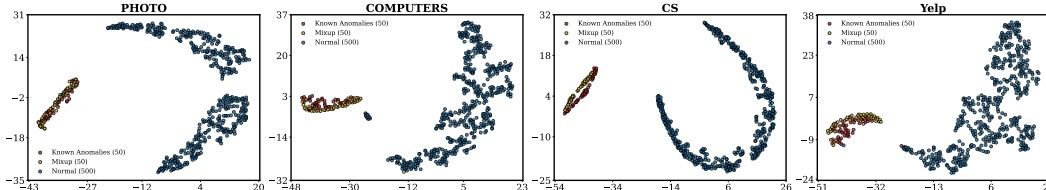

Figure 6: Visualization of sample embeddings after Mixup

detection. Specifically, we vary the loss weight $\lambda_{un} \in \{0.1, 0.3, 0.5, 0.7\}$ and $\lambda_{eng} \in \{0.1, 0.3, 0.5\}$. The results are shown in Figure 7 and 8. Notably, although multi-sample mixup enhances anomaly representation diversity, an overly large mixup ratio may introduce distributional shifts or generate overly blurred features, thereby disrupting the learning of original decision boundaries. This issue becomes more pronounced when labeled anomalies are scarce, often leading to unstable training. To mitigate this, we treat mixup as a lightweight augmentation strategy to provide auxiliary distributional support while minimizing structural disruption, and fix its weight at 0.1. As shown in the Figure, on the Photo dataset, the model achieves optimal performance when the $\lambda_{un} = 0.5$ and the $\lambda_{eng} = 0.1$. For the Computers dataset, the best results are observed when $\lambda_{un} = 0.1$ and $\lambda_{eng} = 0.5$. In summary, the optimal loss weight configuration varies across datasets, which can be attributed to differences in graph structure complexity, anomaly separability, and feature distribution. For example, datasets with more entangled representations or higher intra-class variability (*e.g.*, Computers) may benefit from stronger sample reweighting to focus on informative nodes. Finally, Table 7 shows the loss weight values across the different datasets.

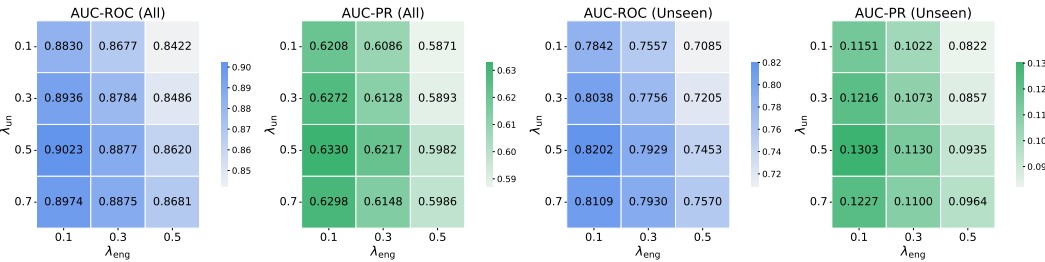

Figure 7: Performance impact of loss weights on the Photo dataset.

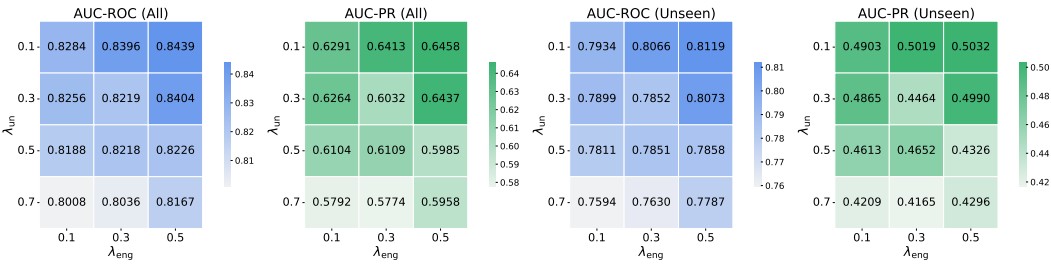

Figure 8: Performance impact of loss weights on the Computers dataset.

## G.5 VISUALIZATION OF MIXUP EMBEDDINGS

To investigate the distribution of synthesized anomalies relative to both seen anomalies and normal data, we conducted an extended t-SNE visualization experiment across datasets with varying characteristics (Photo, Computers, CS, and Yelp). We projected the embeddings of normal nodes, original "Seen Anomaly" samples, and the synthetic "Mixup Anomaly" samples generated by our

method. As illustrated in Figure 6, the visualization reveals two key observations. First, the "Mixup Anomaly" embeddings (yellow) do not merely overlap with the "Seen Anomaly" cluster (red) but form a broader, diffused distribution that expands outward. This confirms that our strategy effectively explores the potential feature space beyond the limited training samples. Second, across all datasets, there remains a clear and distinct separation between the synthesized Mixup anomalies and the normal node clusters. This empirical evidence demonstrates that `DEMO` successfully expands the anomaly decision boundary to enhance generalization without encroaching upon the normal data distribution, thereby ensuring both the diversity of anomaly representations and the stability of model training.

