# OpenReview forum: "Dynamic Multi-sample Mixup with Gradient Exploration for Open-set Graph Anomaly Detection"
_ICLR.cc/2026/Conference — ICLR 2026 Poster_

### Official Review · Reviewer_1orP · 2025-10-18

**Soundness:** 2
**Presentation:** 3
**Contribution:** 2
**Rating:** 4
**Confidence:** 4

**Summary:**

The authors propose an approach named Dynamic Multi-sample Mixup with Gradient Exploration (DEMO) for open-set graph anomaly detection, leveraging a dynamic framework to adapt the optimization procedure with generalizability. Given experimental comparison show the effectiveness to some extent.

**Strengths:**

1. The authors propose an approach named Dynamic Multi-sample Mixup with Gradient Exploration (DEMO) for open-set graph anomaly detection.
2. Given experimental comparison show the effectiveness to some extent.

**Weaknesses:**

1. It can be a question that, since there exists anomaly detection dataset benchmark such as [1], why the authors didn't conduct experiments on the real-world datasets. It will be better to report the experimental results on those datasets.
2. The unseen anomaly simulation can be an significant part of the framework. Therefore, the authors should show the generated embedding of the simulation to show the effectiveness (on above mentioned datasets).
3. The pseudo-labling generation can be an important component of the framework, and the corresponding loss require a high-quality pseudo-labels. The auhors should present if the generated pseudo labels match with the ground truths (on above mentioned datasets).

[1] Jianheng Tang, Fengrui Hua, Ziqi Gao, Peilin Zhao, Jia Li. GADBench: Revisiting and Benchmarking Supervised Graph Anomaly Detection. NeurIPS 2023.

**Questions:**

Please refer to the weaknesses.

---

> ### Author Response · Authors · 2025-11-17
> **Response to the comments**
>
> We greatly appreciate your review and will respond to each of your comments below. Additionally, we have uploaded the revised PDF version, and the corresponding modifications in the text have been marked in blue font.
>
> **Weakness 1: It can be a question that, since there exists anomaly detection dataset benchmark such as [1], why the authors didn't conduct experiments on the real-world datasets. It will be better to report the experimental results on those datasets.**
>
> **Answer:** Thank you for your valuable feedback. We highly recognize GADBench as an excellent GAD benchmark framework, and we have also cited it in our paper. However, we did not adopt all the datasets it includes due to the following key considerations: **(1) The task studied in this paper is Open-Set GAD, which requires datasets to have original multi-class labels.** This is necessary to partition anomalies into "seen" and "unseen" subsets. The datasets in GADBench, while convenient, are mostly pre-processed with binary ('normal' vs. 'anomaly') labels, which does not fit our open-set task definition. **(2) The 6 datasets we adopted are also standard, real-world benchmarks** (e.g., ogbn-arxiv and ogbn-mag). Notably, the Yelp dataset we used, while similar in origin to YelpChi in GADBench, was specifically processed by us to leverage its original multi-class information to create the "multi-anomaly" open-set scenario. **In summary, due to the specific nature of our task, we could not validate our method on standard binary anomaly detection datasets.** Therefore, we adopted equally popular, real-world datasets that meet our multi-class requirements.
>
> **Weakness 2: The unseen anomaly simulation can be an significant part of the framework. Therefore, the authors should show the generated embedding of the simulation to show the effectiveness (on above mentioned datasets).**
>
> **Answer:** Thank you for your valuable suggestion. We will provide a visualization for the embeddings after Mixup as you mentioned.
> **(1) Regarding the Datasets:** First, concerning the "on above mentioned datasets" (i.e., GADBench), we would like to clarify that due to the specific nature of our open-set task, we cannot conduct this experiment on those datasets.
> **(2) Added Experiment:** Therefore, we conducted this visualization experiment on our own multi-class datasets. The result has been added to the Appendix G.5 (as shown in Figure 8). This figure uses t-SNE to visualize the low-dimensional embeddings of the 'seen' anomaly samples from the training set and the corresponding 'Mixup' samples. As can be seen, the embeddings of the Mixup samples do not simply overlap with the 'seen' anomalies. Instead, they diffuse and expand outwards from the 'seen' anomaly regions, effectively bridging the gaps between 'seen' and potential 'unseen' anomaly areas. This experiment demonstrates that the Mixup-generated embeddings successfully expand the decision boundary, forcing the model to learn a more generalized concept of anomaly.
>
>
> **Weakness 3: The pseudo-labling generation can be an important component of the framework, and the corresponding loss require a high-quality pseudo-labels. The auhors should present if the generated pseudo labels match with the ground truths (on above mentioned datasets).**
>
> **Answer:** Thank you for your valuable comment. We have followed your suggestion and validated the quality of our generated pseudo-labels on three datasets (It is worth noting that due to the specific nature of our **open-set** task, we could not perform this validation on the GADBench datasets you previously mentioned). We use the "Match Rate" metric, which represents the percentage of generated pseudo-labels that match the ground truth. **The results of this new experiment (shown below) demonstrate that the quality of pseudo-labels generated by our method (Section 3.4) is very high, achieving a Match Rate of over 92% on all three datasets.**
>
> | Dataset    | Photo  | Computer | CS     |
> | ---------- | ------ | -------- | ------ |
> | Match Rate | 95.83% | 92.31%   | 94.15% |

---

> > ### Comment · Reviewer_1orP · 2025-11-28
> >
> > Thanks for the rebuttal of the authors. However, for weakness 1, I still have a question about why a binary dataset can not be divided into "seen" and "unseen" subsets. Since the typical supervised strategy requires dividing the datasets into training and test sets, why can't the training set be considered as "seen" subsets and the test set as "unseen" subsets? Is it possible to implement in such a setting using other techniques? According to [1], it is possible to conduct such experiments. About weakness 2, could the authors provide the encoded representation of normal nodes as well? Although the known anomalies and Mixup can share a similar pattern in PHOTO, they are not that similar in COMPUTERS and CS, which raises questions: will the encoded representation of normal nodes share a similar pattern with them? will the differences between anomalies and Mixup lead to unstable training for datasets like COMPUTERS and CS? How about the embedding on larger datasets? will they mix together due to the large number? If so, under such a scenario, how would the simulation work to boost the performance?
> >
> > 1. Qizhou Wang, Guansong Pang, Mahsa Salehi, Xiaokun Xia, Christopher Leckie. Open-Set Graph Anomaly Detection via Normal Structure Regularisation. ICLR 2025.

---

> > > ### Author Response · Authors · 2025-11-29
> > >
> > > Thank you for your response. We will address your two questions separately.
> > >
> > > **Response to Weakness 1:** Regarding the binary datasets you mentioned, while they can indeed be divided into "seen" and "unseen" subsets, this does not satisfy our open-set task setting.
> > >
> > > **(1) The distinction between "unseen subsets" and "unseen anomaly types":** In a standard train/test split, although the test set is technically "unseen," it typically belongs to the same anomaly class (i.e., the same distribution) as the training set. **The core challenge of Open-Set GAD is to detect entirely new anomaly types (i.e., distribution shift)**, which is significantly harder than merely detecting new samples of the same class. Therefore, the former belongs to "unseen subsets," whereas the latter is the "unseen anomaly types" required for our task.
> > >
> > > **(2) Binary datasets cannot "isolate" new types:** In binary datasets, all anomalies are labeled as "1". If we split the data randomly, both the training and test sets will represent the same anomaly behavior. In a real open-set setting, we require original multi-class labels (e.g., Anomaly Class A and Anomaly Class B) to ensure the model is trained on Class A and tested on Class B, thereby guaranteeing that the test set represents a truly "new type."
> > >
> > > **(3) Implementation following Work [1]:** Our dataset partition setting strictly follows the work [1] (NSReg) you mentioned, which is explicitly stated in line 320 (Section 4.1) of the main text. Additionally, we have also compared our method with NSReg.
> > >
> > >
> > >
> > > **Response to Weakness 2:** We have addressed this by visualizing the encoded representations of normal nodes in **Appendix G.5 (Figure 6)**. Specifically addressing your concern about **"similar patterns"**:
> > >
> > > Our results on **Computers** and **CS** confirm that while the synthesized Mixup samples exhibit higher variance and are less similar to the original 'Seen Anomalies' compared to the PHOTO dataset, **they do not share similar patterns with 'Normal Nodes'**. As shown in Figure 6, there remains a clear and distinct separation between the Mixup samples (yellow) and the normal cluster (blue). This indicates that the Mixup samples occupy the feature space outside the normal distribution, effectively expanding the anomaly boundary without encroaching on the normal class. This distinct separation ensures that the training remains stable and that the simulation effectively boosts performance by exploring the potential anomaly space, even in large-scale scenarios like **Yelp**.
> > >
> > > [1] Qizhou Wang, Guansong Pang, Mahsa Salehi, Xiaokun Xia, Christopher Leckie. Open-Set Graph Anomaly Detection via Normal Structure Regularisation. ICLR 2025.

---

### Official Review · Reviewer_5VDR · 2025-10-31

**Soundness:** 3
**Presentation:** 2
**Contribution:** 2
**Rating:** 6
**Confidence:** 3

**Summary:**

This paper introduces Dynamic Multi-sample Mixup with Gradient Exploration (DEMO) to tackle open-set graph anomaly detection. Its method (1) synthesizes anomalies via multi-sample mixup at the embedding level, (2) weights samples using validation-gradient signals, and (3) adapts class thresholds with a memory bank. Experiments cover six datasets and twelve baselines with ablation studies and sensitivity analysis.

**Strengths:**

1. Open-set graph anomaly detection is an important problem, and is under-explored.
2. Multi-sample mixup at the embedding level is a neat way to approximate unseen anomalies.
3. Comprehensive experiments covering 6 datasets and 12 baselines with ablations study and sensitivity analysis.

**Weaknesses:**

1. The use of index i seems inconsistent. In Definition 3.1, i appears to index synthetic representations, but in Equation 1, i seems to refer to original seen anomaly nodes. Please clarify the indexing scheme and distinguish between source nodes and synthetic outputs.

2. In Section 3.3, some implementation details are missing for the Hessian matrix computation: (i) How is the Hessian computed in practice? (ii) What are the "optimal parameters"—current parameters at each step or from separate optimization? (iii) Computing and inverting the full Hessian is computationally prohibitive for large weights. Is any approximation like diagonal Hessian or Hessian-vector products used? Implementation details and computational feasibility discussion are needed.

3. The gradient-based weighting in Section 3.3 uses validation loss gradients to compute training sample weights, which appears to use validation data to guide training. This could have the risk of data leakage and overfitting to the validation set. It would be helpful to discuss whether this leads to data leakage or overfitting and, if feasible, include empirical evidence.


4. Section 3.4 would benefit from further clarification. Could you please specify what is meant by “samples selected for class c”? Are these unlabeled samples assigned the pseudo-label c? Given that Table 3 shows pseudo-labeling has a substantial impact on performance, including visualizations of how the thresholds evolve during training could help illustrate this component’s contribution.


5. The manuscript does not include a computational or memory complexity analysis. Please provide the method’s time complexity and memory overhead. Given the use of Hessian matrices and their inverses, a scalability analysis with respect to model sizes is also needed.


6. In Figure 2, similar colors make methods difficult to distinguish. Please use distinct colors for each method.

**Questions:**

See Weakness.

---

> ### Author Response · Authors · 2025-11-17
> **Response to the comments**
>
> We greatly appreciate your review and will respond to each of your comments below. Additionally, we have uploaded the revised PDF version, and the corresponding modifications in the text have been marked in blue font.
>
> **Weakness 1: The use of index i seems inconsistent.**
>
> **Answer:** Thank you for pointing out this important issue with the symbol inconsistency. The symbol `i`  is used as an index for both the input ($z_i^{\text{train}}$) and output ($\hat{z}\_i$), which indeed causes confusion. Therefore, we have clarified this relationship in **Definition 3.1** in the revised version, explicitly stating that each original anomaly ($z_i^{\text{train}}$) generates a corresponding synthetic representation ($\hat{z}_i$). Notably, the index `i` represents the same entity in both places and is not reused. **Definition 3.1** has been modified as follows:
>
> > Given the seen anomaly node set $\mathcal{V}\_a^{\text{train}}=\\{v_1^{\text{train}},v_2^{\text{train}},\cdots,v_N^{\text{train}}\\}$, which are encoded into embeddings $\mathcal{\textbf{Z}}\_a^{\text{train}}=\\{z_1^{\text{train}},z_2^{\text{train}},\cdots,z_N^{\text{train}}\\}$. For each original anomaly $z_i^{\text{train}} \in \mathcal{\textbf{Z}}\_a^{\text{train}}$, we generate a corresponding synthetic representation $\hat{z}\_i$. This synthetic sample is defined as a dynamic mix of all seen anomalies' embeddings: $\hat{z}\_i=\sum\_{j=1}^N\alpha\_{ij}z_j^{\text{train}}$, where $\sum\_{j=1}^N\alpha\_{ij}=1$ and $\alpha_{ij}\in[0,1]$.
>
> **Weakness 2: Some implementation details of Hessian matrix**
>
> **Answer:** Thank you for your comment. We will respond to each point: **(1)** In practice, we **do not** compute or store the full Hessian matrix. Instead, we employ an efficient **Diagonal Hessian Approximation**. We use two backpropagation passes to efficiently extract the diagonal elements of the Hessian and accumulate them into a diagonal vector. **(2) The "optimal parameters" refer to the model's parameters at the current training step** when this function is called. Before calculating the influence scores, we set the model to evaluation mode to use the parameters from the current step. **(3) Yes, we explicitly use an approximation**, which is key to our method's computational feasibility. We only use the diagonal of the Hessian and perform an **element-wise inversion** by adding a damping term. Therefore, the final influence score calculation is just an efficient element-wise product and a dot product, completely avoiding expensive matrix inversion.
>
> **Weakness 3: Validation data have the risk of data leakage and overfitting**
>
> **Answer:** Thank you very much for your valuable comments.
> **First**, as shown in **Equation (8)**, this influence score $\beta\_{v_i}$ does **not** re-weight the main classification loss ($\mathcal{L}(v\_{i},y\_{i};\theta)$), but **only** modulates the auxiliary **"energy guidance term"** ($E\_{\theta}(v_{i})$). **Our goal is not to 'overfit' the validation data, but to use the validation set as a 'probe' to dynamically identify the most critical, uncertain, or boundary samples in the training set that are key for generalization. Secondly,** the validation set is processed separately during the data splitting, and our final results are evaluated on a dedicated test set. **Therefore, this validation step does not affect the final results.** Specifically, as shown in the ablation experiments, when the gradient exploration module is removed, the performance drops, which demonstrates the effectiveness of this strategy.
>
> **Weakness 4: Section 3.4 would benefit from further clarification.**
>
> **Answer:** Thank you for your comment. (1)  The term "**samples selected for class c**" indeed refers to the unlabeled samples assigned the pseudo-label **c**. (2) Based on your suggestion, we have added a "visualization experiment" in **Appendix G.6 (as shown in Figure 8)** to illustrate the evolution of the pseudo-labeling thresholds during training. This chart tracks the changes in the anomaly-class threshold ($\tau\_{t}^{+}$) and the normal-class threshold ($\tau\_{t}^{-}$). As shown in the figure, the results exhibit a clear trend: $\tau\_{t}^{+}$ progressively **increases** (towards 1.0), while $\tau_{t}^{-}$ progressively **decreases** (towards 0.0). This visualization empirically confirms our claim in **Section 3.4**, demonstrating that our memory bank-guided strategy effectively adapts as the model's confidence improves, ensuring the pseudo-labeling selection criteria become increasingly stringent.

---

> ### Author Response · Authors · 2025-11-17
> **Response to the comments2**
>
> **Weakness 5: The manuscript does not include a computational or memory complexity analysis. Please provide the method’s time complexity and memory overhead. Given the use of Hessian matrices and their inverses, a scalability analysis with respect to model sizes is also needed.**
>
> **Answer:** We summarize the complexity analysis of the entire **DEMO** framework as follows:
>
> **(1) Time Complexity**: Our method consists of the following three modules:
>
> - The **Mixup module**, which primarily handles few seen anomaly samples, has a complexity of $O(N_a)$ (where $N_a$ is typically 50 in our experiments).
> - The **PL module**, which handles unlabeled samples, has a complexity of $O(N_{\text{unlabeled}})$.
> - The **EG module**, which requires computing the **Hessian matrix**. However, since we use **diagonal Hessian approximation**, the computational cost is significantly reduced, and the complexity is mainly dependent on the number of training and validation samples, i.e., $O(N\_{\text{train}} + N_{\text{val}})$.
>
> **(2) Space Complexity**: In terms of memory storage, the primary memory overhead comes from the **EG module**, but it only requires storing the **diagonal elements of the Hessian** and the **gradients of the validation set**, which are vectors of size equal to the number of model parameters $M$ (72.6M in our experiments). Thus, the additional space complexity introduced by our method is $O(M)$.
>
> **Weakness 6: In Figure 2, similar colors make methods difficult to distinguish. Please use distinct colors for each method.**
>
> **Answer:** Thank you for your valuable comment. We have replaced the colors in Figure 2 with more distinct ones in the revised version.

---

### Official Review · Reviewer_5pMP · 2025-11-02

**Soundness:** 2
**Presentation:** 3
**Contribution:** 2
**Rating:** 6
**Confidence:** 4

**Summary:**

The paper studies open-set GAD, where the goal is to detect both seen and unseen anomalies with limited labelled data. It proposes DEMO, a dynamic multi-sample mixup and gradient exploration framework that generates diverse anomaly samples and adaptively re-weights them based on energy gradients. A memory-bank pseudo-labelling module is further introduced to enhance training stability and handle label scarcity. The perposed method on avergae outperform the baseline methods on the selected datasets.

**Strengths:**

1. The paper explores an interesting and important setting of open-set graph anomaly detection.
2.  The paper is well-written and easy to follow, with clear structure and presentation.
3. Theoretical analysis is provided for the proposed augmentation methods for GAD.

**Weaknesses:**

1. The proposed method relies on data augmentation (multi-sample mixup) as a key contribution, but none of the baseline methods use comparable augmentation strategies. If augmentation is the main source of improvement, the paper should include comparisons with existing augmentation-based methods such as GraphSMOTE (Zhao et al. 2021) or other graph Mixup approaches to ensure a fair evaluation.

2. The ablation results show that removing the pseudo-labelling component leads to a notable performance drop. Does this mean much of the improvement may come from this module rather than the proposed mixup strategy?   It would be helpful for the authors to further clarify on this.

2. The GAD-related work section fails to mention recent methods that leverage label information, such as meta-learning GAD (Meta-GDN, Ding et al., 2021), cross-domain GAD (ACT, Wang et al., 2023), and generalist GAD (ARC, Liu et al., 2024; AnomalyGMF, Qiao et al., 2024).

3. Since there are two hyperparameters in the main loss, the sensitivity study should cover all datasets and include comparisons with the baselines. The results in Tables 6 and 7 also appear quite sensitive to these parameters.

4. Several references cited as related work are already accepted by recent conferences but are still listed in arXiv format.

5. The number of supervised baselines and datasets used in the experiments is relatively limited. A good reference for finding additional datasets and resources is the GAD Benchmark (Tang et al., 2023).

**Questions:**

Please refer to my weaknesses.

---

> ### Author Response · Authors · 2025-11-17
> **Response to the comments**
>
> We greatly appreciate your review and will respond to each of your comments below. Additionally, we have uploaded the revised PDF version, and the corresponding modifications in the text have been marked in blue font.
>
> **Weakness 1: Compare with augmentation-based methods like GraphSMOTE for fair evaluation.**
>
> **Answer:** Thank you for your valuable comments. We have included GraphSMOTE and the other algorithms mentioned in **Weakness 6** as additional baselines for comparison. All algorithms were evaluated in the Open-Set setting, where training was conducted using data from the normal class and one anomalous class, with testing extended to other anomaly classes. **The experimental results, as shown below, indicate that although GraphSMOTE demonstrates superior performance in the open-set scenario, our method still achieves SOTA performance.**
>
> |Dataset| Photo (AUC-ROC/AUC-PR) |Computers (AUC-ROC/AUC-PR)|
> |----------|----------------------|--------------------------|
> |GAT| 0.6684/0.1694|0.4519/0.1061|
> |GraphSAGE| 0.5967/0.1091|0.4903/0.1306|
> |PCGNN| 0.6919/0.3261| 0.5617/0.2938|
> |GraphSMOTE| 0.8359/0.5731|0.6538/0.3764|
> |DEMO| **0.9023/0.6330**|**0.8439/0.6458**|
>
> **Weakness 2: Clarify that pseudo-labelling and mixup complement each other, not replace.**
>
> **Answer:** Thank you for your valuable comments. As you pointed out, in the ablation experiments, the performance of our method drops significantly when the pseudo-labelling component is removed. **However, this does not imply that the Mixup or Gradient Exploration (EG) components are less important. We believe that Mixup and EG are the foundational components, while pseudo-labelling is an additional mechanism that enhances performance, making its removal lead to a more significant drop in performance.** Specifically, we observe that the removal of any component results in a performance drop, and it is the collaborative interaction between these components that achieves optimal performance. Furthermore, the effectiveness of the pseudo-labelling component is closely tied to Mixup and Gradient Exploration. High-quality pseudo-labelling relies on the model’s ability to distinguish between normal and anomalous samples, and Mixup "simulates" unseen anomalies, thereby expanding the model's "cognitive boundary" for anomalies. This improvement helps enhance the accuracy of the labelling process for unlabeled samples, which is why the removal of the pseudo-labelling component causes a larger performance decline. Similarly, the Gradient Exploration module also plays a supportive role by enhancing the quality of pseudo-labelling.
>
> **Weakness 3: Add related works.**
>
> **Answer:** Thank you for your suggestion regarding the related work. We have carefully reviewed the methods you mentioned and have included references to Meta-GDN (Ding et al., 2021), ACT (Wang et al., 2023), ARC (Liu et al., 2024), and AnomalyGMF (Qiao et al., 2024) in the GAD part of Section 2.
>
> **Weakness 4: More sensitivity study**
>
> **Answer:** Please refer to the other reply.
>
> **Weakness 5: Several references cited as related work are already accepted by recent conferences but are still listed in arXiv format.**
>
> **Answer:** Thank you for pointing that out. We have updated the references for the papers that have been accepted.
>
> **Weakness 6: The number of supervised baselines and datasets used in the experiments is relatively limited. A good reference for finding additional datasets and resources is the GAD Benchmark (Tang et al., 2023).**
>
> **Answer:** Thank you for your valuable feedback. We would like to take this opportunity to clarify our considerations regarding GADBench:
>
> **(1) Regarding the Dataset Mismatch:** We highly recognize GADBench as an excellent benchmark. However, it primarily targets the **supervised binary classification** (Normal vs. Anomaly) GAD task. Our paper studies the **Open-Set** scenario. This specific setting **requires** datasets with original **multi-class** labels, enabling us to partition anomalies into "seen" and "unseen" subsets. The datasets in GADBench, which are pre-processed for binary classification (with only one anomaly class), cannot be adapted for our open-set task requirements.
>
> **(2) Regarding Added Supervised Baselines:** We have adopted your suggestion and selected 3 representative **supervised models** from the GADBench framework (e.g., GAT, GraphSAGE) for comparison. To ensure fairness, these supervised models were also run under our **open-set protocol**. As shown in our response to **Weakness 1** (and its corresponding table), our method still achieves SOTA performance under this fair comparison.

---

> ### Author Response · Authors · 2025-11-17
> **Response to the comments2**
>
> **Weakness 4: Since there are two hyperparameters in the main loss, the sensitivity study should cover all datasets and include comparisons with the baselines. The results in Tables 6 and 7 also appear quite sensitive to these parameters.**
>
> **Answer:** Thank you for your valuable comments. We have performed a sensitivity analysis of the hyperparameters in **Appendix G.4**, and the experimental results on the **Photo** and **Computers** datasets are shown in Figures **6** and **7**. Based on your suggestion, we will extend this analysis to the remaining four datasets. As shown in the results below, the two parameters following **DEMO** represent $\lambda_{un}$ and $\lambda_{eng}$. The results indicate that our method still outperforms the baselines in most cases. Additionally, the analysis shows that for multiple datasets, performance under different hyperparameter configurations is superior to the baselines. Therefore, while our method is sensitive to these hyperparameters, on most datasets (especially small-scale datasets), any combination of the hyperparameters achieves **SOTA** performance.
>
>
>
> | Dataset        | CS (AUC-ROC/AUC-PR) | Yelp (AUC-ROC/AUC-PR) | ogbn-arxiv (AUC-ROC/AUC-PR) | ogbn-mag (AUC-ROC/AUC-PR) |
> | -------------- | ------------------- | --------------------- | --------------------------- | ------------------------- |
> | ConsisGAD      | 0.7178/0.5271       | 0.6988/0.2970         | 0.6216/0.3148               | 0.4909/0.0043             |
> | SpaceGNN       | 0.7784/0.6587       | 0.6853/0.2916         | 0.6133/0.3301               | 0.4626/0.0043             |
> | NSReg          | 0.9032/0.8115       | 0.7015/**0.3029**     | 0.6182/0.3230               | 0.4836/0.0041             |
> | DEMO (0.1-0.1) | 0.9273/0.8678       | 0.6958/0.1882         | 0.6343/0.3295               | 0.4874/0.0051             |
> | DEMO (0.1-0.3) | 0.9208/0.8664       | 0.6655/0.1997         | 0.5715/0.2457               | 0.4852/0.0049             |
> | DEMO (0.1-0.5) | 0.9335/0.8683       | 0.6767/0.2019         | 0.5668/0.2422               | 0.4754/0.0049             |
> | DEMO (0.3-0.1) | 0.9429/0.8836       | **0.7097**/0.2238     | 0.6338/0.3298               | 0.4961/0.0054             |
> | DEMO (0.3-0.3) | **0.9448/0.8857**   | 0.6749/0.2145         | 0.6107/0.3001               | 0.4908/0.0053             |
> | DEMO (0.3-0.5) | 0.9316/0.8522       | 0.6864/0.2149         | 0.5686/0.2434               | 0.4919/0.0053             |
> | DEMO (0.5-0.1) | 0.9299/0.8614       | 0.6343/0.1905         | **0.6364/0.3329**           | **0.4967/0.0054**         |
> | DEMO (0.5-0.3) | 0.9325/0.8648       | 0.6584/0.1841         | 0.6177/0.3127               | 0.4851/0.0054             |
> | DEMO (0.5-0.5) | 0.9304/0.8543       | 0.6505/0.1908         | 0.5889/0.2644               | 0.4939/0.0052             |

---

### Official Review · Reviewer_PTtY · 2025-11-04

**Soundness:** 2
**Presentation:** 3
**Contribution:** 2
**Rating:** 4
**Confidence:** 4

**Summary:**

The authors propose a framework, DEMO (Dynamic Multi-sample Mixup with Gradient Exploration), to address the problem of open-set graph anomaly detection (GAD). This task aims to detect not only anomalies seen during training but also novel, unseen types of anomalies, using only a small set of labeled data. The core idea is a three-part dynamic training framework. First, to generalize from limited seen anomalies, DEMO uses a multi-sample mixup strategy that adaptively fuses multiple seen anomaly nodes to synthesize new, diverse samples, thereby expanding the decision boundary. Second, it employs an energy gradient-driven feedback mechanism to dynamically re-weight all training samples, prioritizing uncertain and informative nodes to ensure a more robust optimization process. Third, to combat label scarcity and class imbalance, it maintains a memory bank of historical predictions to guide a pseudo-labeling process with adaptive, class-specific confidence thresholds. The authors demonstrate that DEMO achieves performance on six real-world graph datasets under various open-set evaluation settings.

**Strengths:**

There are a few things I like about the paper:
1. The paper addresses a practical challenge in graph anomaly detection, moving from the closed-set assumption to a more realistic open-set scenario. This is relevant in many application areas when the anomalies are always evolving.
2. The authors provide experimental validation on six benchmark datasets of varying scales and domains.
3. The authors provide theoretical analysis of the proposed method.
4. The authors performed an ablation study to see the source of the performance improvement.

**Weaknesses:**

1. The work is motivated by open set problem setting. Which aims to also detect new types of anomalies. However, the main ingredient of the proposed method is mixup, where it can only detect anomalies that are linear combinations of seen anomalies in the embedding space. It will be less effective in detecting totally novel anomalies outside the one that have been seen before (which is the advantage of a purely unsupervised model).
2. There are two types of anomalies that need to be considered in the evaluation, anomalies that come from existing sets/classes, and anomalies that come from new set/classes (open-set). The paper, however, does not distinguish these two in the evaluation. I would suggest the authors add more experimentation that distinguishes these two. Particularly, I am interested in the performance of the proposed method vs semi-supervised baselines on detecting anomalies from existing sets/classes, and the performance of the proposed method vs purely unsupervised baselines on detecting anomalies from new sets/classes.
3. In the large-scale experiments, DEMO's AUC-PR score on the Yelp dataset is noticeably lower than the best-performing baseline (0.2238 vs. NSReg's 0.3029). Given that AUC-PR is a critical metric for highly imbalanced datasets, this specific underperformance needs further exploration.
4. The adaptation of purely unsupervised models into semi-supervised models is not clearly explained in the paper.

**Questions:**

Please address the weaknesses mentioned above.

---

> ### Author Response · Authors · 2025-11-17
> **Response to the comments**
>
> We greatly appreciate your review and will respond to each of your comments below.
>
> **Weakness 1: Mixup detects anomalies as combinations of seen ones, limiting novel anomaly detection.**
>
> **Answer:** Thank you for this insightful comment. We would like to clarify this from three perspectives: **(1)** We acknowledge that our method creates linear combinations of seen anomalies in the embedding space. **Its purpose is to prevent the model from overfitting to the scarce 'seen' anomalies and to force the model to learn a broader and more robust decision boundary**, thereby enhancing its ability to detect 'unseen' anomalies. **(2) The generalization capability of the proposed method does not rely solely on Mixup, but on a synergistic framework**. For instance, the Gradient Exploration (EG) mechanism prioritizes the challenging samples generated by Mixup that lie on the decision boundary, to achieve a more robust optimization. **(3)** Finally, our experimental evidence (e.g., Tables 5 and 6 in the Appendix) demonstrates that **DEMO's performance in detecting 'unseen' anomalies is significantly superior to all baselines, including purely unsupervised methods**.
>
> **Weakness 2: Distinguish between known and unseen anomalies in evaluation, add comparisons.**
>
> **Answer:** Thank you for your valuable feedback. We fully agree on the importance of distinguishing the evaluation of 'seen' and 'unseen' anomalies. **(1) Regarding 'unseen' anomalies (i.e., your suggested Experiment 2):** We would like to clarify that this experiment was already reported in the original manuscript in **Appendix G.1 (Tables 5 and 6)**. As described, this experiment evaluates performance only on the 'unseen' anomalies and compares DEMO against all baselines (including the unsupervised ones you mentioned). The results confirm that DEMO significantly outperforms all competitors in this pure open-set scenario. **(2) Regarding 'seen' anomalies (i.e., your suggested Experiment 1):** We have adopted your suggestion and **added a new experiment** (see table below). This experiment specifically compares DEMO against 3 recent SOTA semi-supervised methods on their performance at detecting only the 'seen' anomalies. The results show that our method achieves SOTA performance in this closed-set scenario as well.
>
> |Datasets|Photo (AUC-ROC/AUC-PR)|Computers (AUC-ROC/AUC-PR)|CS (AUC-ROC/AUC-PR)|
> |---------|----------------------| -------------------------- | ------------------- |
> |ConsisGAD|0.9537/0.9112|0.8531/0.7952|0.8952/0.8132|
> |SpaceGNN|0.9206/0.8773|0.9701/0.9003|0.9715/0.9061|
> |NsReg|0.9224/0.8528|0.8845/0.8287|0.9690/ 0.9247|
> |DEMO|**0.9965/0.9712**|**0.9872/0.9233**|**0.9984 / 0.9874**|
>
> **Weakness 3: Investigate AUC-PR drop on Yelp, explain performance gap with baselines.**
>
> **Answer:** Thank you for your valuable observation. We acknowledge that DEMO's AUC-PR on Yelp (Table 2) is lower than NSReg's. The following is our analysis of the phenomenon.
>
> **(1) Different Strategies:** NSReg's core is to "tighten the normal class" boundary, whereas DEMO's core is to "expand the anomaly class" boundary (via Mixup) to cover potential unseen classes. NSReg's strategy is highly precise for 'seen' anomalies, leading to a high AUC-PR in the mixed test set of Table 2, which contains many 'seen' anomalies. In contrast, DEMO's strategy, which "blurs" the 'seen' anomaly boundary to achieve generalization, sacrifices precision on the 'seen' portion in Table 2.
>
> **(2) Evidence on 'Unseen' Classes:** Table 6 in Appendix G.1 (evaluating only 'unseen' anomalies) reveals the fundamental difference. On Yelp, NSReg's performance collapses (AUC-PR drops from 0.3029 to 0.0178), proving its high score was dependent on 'seen' anomalies. DEMO, however, performs best on the 'unseen' category (AUC-PR 0.0635, AUC-ROC 0.7235). This demonstrates that DEMO's strategy is superior in achieving the core goal of open-set generalization.
>
> **Weakness 4: The adaptation of purely unsupervised models into semi-supervised models is not clearly explained in the paper.**
>
> **Answer:** Thank you for this comment. We realize this explanation may not have been prominent enough in the main text. We did mention in Section 4.1 (**Baselines**) that we adapted four unsupervised models (TAM, OCGNN, DOMINANT, and AnomalyDAE) to fit the semi-supervised setting. The detailed adaptation strategy is provided in the **first paragraph of Appendix E**. The specific operations are as follows: **(1) For TAM**, we refine its affinity maximization objective to focus exclusively on labeled normal and partial abnoraml nodes. **(2) For OCGNN**, the one-class center optimization is constrained to labeled normal and partial abnoraml instances. **(3) For DOMINANT and AnomalyDAE** (which are reconstruction models), we restrict their auto-encoding loss computation to labeled normal and partial abnoraml nodes during training.

---

> > ### Comment · Reviewer_PTtY · 2025-11-28
> >
> > Thank you to the authors for proving responses to my concerns.
> > It addressed some of my concerns remain.
> >
> > In real world applications, such as fraud detection, many novel anomalies occurs due to the adversary nature of the fraudster.  This kind of anomalies cannot be solved by just performing mixup on known classes. Hence, for these type of anomalies, purely unsupervised learning often perform better than purely supervised learning.
> >
> > I would suggests the authors to design task to accommodates two types.
> > - Anomalies that dominated by known types (known unknown). We can assess by seeing performance of supervised model better than unsupervised models.
> > - Anomalies that dominated by totally unknown types (unknown unknown). We can assess by seeing performance of unsupervised model better than supervised ones.
> >
> > Then, we would like to see how the proposed approach stack in these two types.
> >
> > I do not see the second types of anomalies in the experiment design, even the ones in the appendix.

---

> > > ### Author Response · Authors · 2025-11-28
> > >
> > > Thank you for your response. We hope to clarify that, assuming we have not misunderstood your point, we have actually already covered both types of experiments. Additionally, we need to note that the training set for both experiments consists of normal class samples and one class of anomaly samples, which aligns with the design of the open-set scenario.
> > >
> > > **(1) Experiment Correspondence:**
> > >
> > > - **Type 1 (Known Unknown):** This corresponds to **Table 1 and Table 2** in the main text. The test set includes both 'known' and 'unknown' anomalies. The results show that DEMO outperforms all baselines.
> > > - **Type 2 (Unknown Unknown):** This corresponds to **Table 5 and Table 6 in Appendix G.1**. In this setting, we completely removed all 'known' anomaly classes from the test set. All test anomalies come from entirely new categories not seen during training. This strictly conforms to your definition of "Totally unknown." The results also show that DEMO outperforms other baselines.
> > >
> > > **(2) Explanation and Analysis:** You mentioned that in Type 2 scenarios, unsupervised models usually perform better than supervised models. Our experimental results (Table 5 in Appendix G.1) show that some semi-supervised methods indeed experienced a performance collapse, performing worse than unsupervised methods. However, **DEMO still achieved SOTA. Using the Mixup strategy to simulate some unseen anomalies to improve performance is one reason. Another important reason is that DEMO also incorporates the advantages of unsupervised learning**: our Pseudo-Labeling (PL) module leverages the massive amount of unlabeled data and is dedicated to learning a more precise boundary for the "Normal class." Therefore, even when facing unseen anomalies ("Unknown Unknown" anomalies), as long as they deviate from the tightly defined "Normal" distribution, DEMO can identify them through the tightened normal boundary.
> > >
> > > The above is our response to your question, and we hope we have not misunderstood your meaning. If you have any further doubts, we welcome you to raise them.

---

### Meta-Review · Area_Chair_Ardy · 2026-01-06

**Summary:**

The paper studies open-set graph anomaly detection and proposes DEMO, a dynamic framework using multi-sample mixup, gradient exploration, and pseudo-labeling. Reviewers agreed the problem is important and under-explored.
They raised concerns about whether mixup can capture truly unknown anomalies, the realism of the evaluation protocol, fairness of comparisons, and the role of pseudo-labeling. Through rebuttal, the authors largely address the concerns by adding new experiments, new baselines, detailed analyses, and clarifications.

**Reviewer Concerns:**

Well addressed by rebuttal
- Added experiments separating known-unknown and unknown-unknown anomalies.
- Clear comparisons with augmentation methods (e.g., GraphSMOTE) and supervised baselines.
- Provided visualization of mixup embeddings and pseudo-label quality analysis.
- Added complexity analysis and detailed implementation clarifications.

Remaining but minor

- Evaluation still relies on benchmark-style datasets rather than adversarial real-world deployments.

**Reviewer Scores:**

Reviewer PTtY (rating 4) has a core concern on unknown-unknown anomalies. The concern was addressed by the rebuttal. He is likely to increase the rating to 6.

Reviewers 5pMP and 5VDR are likely to maintain a rainting of 6.

Reviewer 1orP's follow-up questions seem to be addresed by the authors. The rating is likely to increase.

---

### Decision · Program_Chairs · 2026-01-26

Accept (Poster)